# Interplay of mechanics and chemistry governs wear of diamond-like carbon coatings interacting with ZDDP-additivated lubricants

Valentin R. Salinas Ruiz[1,2,3,8], Takuya Kuwahara [4,8], Jules Galipaud[1,2], Karine Masenelli-Varlot [2], Mohamed Ben Hassine[1], Christophe Héau[3], Melissa Stoll[4], Leonhard Mayrhofer[4], Gianpietro Moras [4], Jean Michel Martin[1], Michael Moseler [4,5,6,7✉] & Maria-Isabel de Barros Bouchet [1✉]

Friction and wear reduction by diamond-like carbon (DLC) in automotive applications can be affected by zinc-dialkyldithiophosphate (ZDDP), which is widely used in engine oils. Our experiments show that DLC's tribological behaviour in ZDDP-additivated oils can be optimised by tailoring its stiffness, surface nano-topography and hydrogen content. An optimal combination of ultralow friction and negligible wear is achieved using hydrogen-free tetrahedral amorphous carbon (ta-C) with moderate hardness. Softer coatings exhibit similarly low wear and thin ZDDP-derived patchy tribofilms but higher friction. Conversely, harder ta-Cs undergo severe wear and sub-surface sulphur contamination. Contact-mechanics and quantum-chemical simulations reveal that shear combined with the high local contact pressure caused by the contact stiffness and average surface slope of hard ta-Cs favour ZDDP fragmentation and sulphur release. In absence of hydrogen, this is followed by local surface cold welding and sub-surface mechanical mixing of sulphur resulting in a decrease of yield stress and wear.

---

[1] University of Lyon, Ecole Centrale de Lyon, Laboratory of Tribology and System Dynamics, CNRS UMR5513, Ecully, France. [2] University of Lyon, INSA-Lyon, UCBL, MATEIS UMR CNRS, Villeurbanne, France. [3] HEF/IREIS, Avenue Benoît Fourneyron, Andrézieux-Bouthéon, France. [4] Fraunhofer Institute for Mechanics of Materials IWM, MicroTribology Center µTC, Freiburg, Germany. [5] Cluster of Excellence livMatS, Freiburg Center for Interactive Materials and Bioinspired Technologies, University of Freiburg, Freiburg, Germany. [6] Institute of Physics, University of Freiburg, Freiburg, Germany. [7] Freiburg Materials Research Center, University of Freiburg, Freiburg, Germany. [8] These authors contributed equally: Valentin R. Salinas Ruiz and Takuya Kuwahara. ✉email: michael.moseler@iwm.fraunhofer.de; maria-isabel.de-barros@ec-lyon.fr

Diamond-like carbon is an excellent solid lubricant that exhibits high wear resistance and superlubricity (i.e. friction coefficient $\mu$ below 0.01) under boundary[1,2] and mixed lubrication[3] with organic friction modifiers (FM), including oleic acid and glycerol. It has been increasingly applied as a protective coating in internal combustion engines, where operating conditions are extremely severe and thus the use of anti-wear (AW) additives, e.g. zinc dialkyldithiophosphate (ZDDP) and FM additives, e.g. molybdenum dithiocarbamate (MoDTC), admixed with base oils is essential for reducing wear and friction. However, some additives (e.g. MoDTC[4]) can cause massive wear of some DLCs. Particularly the role of ZDDP in the wear of DLC/DLC contacts remains controversial[5–9] since research on these additives has mainly focused on conventional ferrous surfaces[10]. Consequently, a better understanding of interactions between such additives and DLC as well as an improvement of the AW performance of additives on DLC are technologically important.

ZDDP is one of the most indispensable and effective AW additives for automotive combustion engines[10]. However, since sulphur and phosphorous produce environmentally hazardous waste substances and degrade catalyst converters, alternative additives have been sought to replace ZDDP with other ash- and phosphorous-free AW agents for the sake of reducing the environmental load[11,12]. Nevertheless, owing to its extraordinary AW properties, ZDDP is still a principal AW additive. Substantial efforts have been made to identify chemical structures and formation mechanisms of ZDDP-derived tribofilms[13–16], which is necessary not only to improve the AW performance of ZDDP but also to formulate alternative sulphur- and phosphorous-free lubricants.

Particular attention is paid to whether ZDDP is effective with DLC, which has been already investigated for various types of DLC coatings[5,6,9,17,18], including hydrogen-free tetrahedral amorphous carbon (ta-C) and hydrogenated amorphous carbon (a-C:H). Vengudusamy et al. carried out comprehensive experimental studies on the effect of different types of DLC coatings, such as a-C:H, a-C, ta-C and doped DLCs[5–7]. They reported that ta-C lowers boundary friction, but exhibits higher wear compared with a-C:H and a-C. They concluded that DLC's wear resistance is mainly governed by the type of DLC, but the role of ZDDP as AW agent is unclear.

Interestingly, ZDDP-derived patchy films can form on DLC surfaces, but are less thick and durable than those observed on steel, probably due to weaker interactions of ZDDP with DLC surfaces[7,19]. The thicknesses and morphologies of these tribofilms are affected by intrinsic parameters of the DLC coatings, i.e. hydrogen content, $sp^2/sp^3$ ratio, presence of dopants, surface roughness and mechanical properties[6,20]. The former three factors alter chemical reactivity of DLC surfaces and thus reaction rates of ZDDP, whereas the latter two factors affect the mechanical response of the system such as contact pressure and geometry, and elastic/plastic deformation of surface asperities.

Mosey et al.[21] simulated the cross-linking of bulk zinc phosphate precursors with ab-initio molecular dynamics (MD), showing that high hydrostatic pressures induce the formation of poly-phosphates. It would be tempting to ascribe the low quantity of ZDDP-derived polyphosphates on DLC to reduced contact pressures. However, DLCs are stiffer than steel and therefore ZDDP on DLC should experience higher contact pressures. This indicates that the chemical interactions between ZDDP and surfaces cannot be neglected in ab-initio modelling. Yet, only decomposition of other sulphur-[22] and phosphorous-containing additives[23] on iron surfaces have been simulated so far.

Here, we combine experiments and simulations to understand tribochemical reactions of ZDDP with DLC and the role of DLC's mechanical properties, surface roughness and chemistry. We study friction and wear of five different self-mated DLC coatings sliding in a ZDDP solution with a PAO base oil. First, a combination of sliding tests, surface chemical analyses, and contact-mechanics calculations reveal the crucial role of asperity-scale contact mechanics on wear and friction. Surface asperities on soft DLC experience lower effective contact pressures ($P_{l,eff}$) than asperities on hard DLC. While 20-nm-thick tribo-patches form on the former and no visible wear is observed, the latter undergo severe wear and a drastic friction increase. Surface chemical analyses reveal significantly more S−C than Zn−S bonds on hard DLCs, indicating that ZDDP reacts preferentially with hard DLC coatings. Second, quantum-chemical simulations reveal that the kinetics of ZDDP's chemical-decomposition reactions is accelerated by $P_{l,eff}$ while it is independent of the DLC's bulk chemical structure. However, chemical differences between DLC surfaces alter the shear response of the frictional interface at high $P_{l,eff}$ (>5 GPa). Non-hydrogenated DLC surfaces are likely to cold-weld at high $P_{l,eff}$ owing to lower steric hindrance and strong attractive interactions. The cold-welded surfaces undergo shear-induced chemical mixing, resulting in sub-surface sulphur doping, which in turn weakens the DLC matrix (reduces its yield stress) causing severe wear. In contrast, local contact pressures on soft DLC are small enough ($P_{l,eff} \lesssim 3.5$ GPa) to prevent cold welding and sulphur doping. In this case, ZDDP chemisorbs and decomposes into atoms and small fragments that accumulate and cross-link to form anti-wear tribofilms. This study elucidates how the interplay between inter-asperity nano-mechanics and chemistry governs tribochemistry and macroscopic tribological behaviours of ZDDP/DLC systems. The effective local pressure at asperity contacts and the coatings' hydrogen content are the crucial mechanical and chemical factors, respectively. The former controls ZDDP's mechanochemistry, whereas the latter affects shear accommodation at the sliding interface.

## Results

**Reciprocating sliding tests**. In order to investigate the impact of ZDDP on friction and wear of DLC coatings, reciprocating cylinder-on-disc sliding tests are performed for self-mated DLC/DLC tribopairs in both pure poly-alpha-olefin (PAO-4) base oil and PAO-4 mixed with 1 wt% ZDDP. Five DLC coatings with varying properties are tested under the same conditions (Table 1). They are labelled by a-C:H, a-C, ta-C(51), ta-C(66) and ta-C(78). The values in parenthesis represent the hardness of ta-Cs in GPa. The tests were repeated at least 4 times for each DLC to ensure the repeatability of our experiments. Significant differences in both friction and wear are observed between pure PAO and PAO + ZDDP lubrication. Figure 1a, b show the evolution of friction in pure PAO and PAO + ZDDP, respectively. For a-C and a-C:H, $\mu$ increases gradually and reaches a steady state with $\mu \approx 0.1$ in PAO (Fig. 1a). Conversely, in PAO + ZDDP, the running-in periods are much shorter than those in PAO and $\mu$ decreases slightly to 0.08−0.09 (Fig. 1b). The three ta-Cs reach rapidly a steady, ultralow friction state ($\mu = 0.01 − 0.03$) in PAO. In PAO + ZDDP, the friction curve for ta-C(51) is almost the same as that in PAO. However, for harder ta-Cs, i.e. ta-C(66) and ta-C(78), the addition of ZDDP significantly changes their tribological performance. Friction increases gradually to $\mu = 0.08 − 0.10$ after 2000 and 12000 sliding cycles for ta-C(66) and ta-C(78), respectively.

Figure 1c shows composite wear volumes $W$ measured on discs and cylinders. In PAO, except for ta-C(78), the wear volumes are negligibly small. However, in PAO + ZDDP wear increases drastically for ta-C(66) and ta-C(78) in correlation with the increased $\mu$ (Fig. 1b). Optical images of wear tracks on both cylinders and discs after sliding show deep scratches for ta-C(66)

**Table 1 Chemical compositions, mechanical and structural properties, and surface roughness parameters of the DLC coatings as used in this study.**

| Label | Composition (at%) | | Young's modulus $E$ (GPa) | Hardness $H$ (GPa) | Density $\rho$ (g cm$^{-3}$) | $sp^3$ C $p_{sp^3}$ (%) |
|---|---|---|---|---|---|---|
| | C | H | | | | |
| a-C:H | 80 | 20 | 259 | 27 | 2.07 | - |
| a-C | >99 | <1 | 287 | 24 | 2.34 | 31 |
| ta-C(51) | >99 | <1 | 493 | 51 | 2.77 | 62 |
| ta-C(66) | >99 | <1 | 572 | 66 | 2.91 | 73 |
| ta-C(78) | >99 | <1 | 625 | 78 | 3.01 | 79 |

| Label | RMS after running-in | | | | | |
|---|---|---|---|---|---|---|
| | height $h_{rms}$ (nm) | | | slope $h'_{rms}$ (-) | | |
| | Cylinder | Disc | Composite | Cylinder | Disc | Composite |
| a-C:H | 5.0 ± 0.7 | 6.7 ± 1.3 | 8.4 ± 1.0 | 0.051 ± 0.015 | 0.087 ± 0.043 | 0.104 ± 0.032 |
| a-C | 13.4 ± 3.9 | 26.5 ± 0.1 | 29.8 ± 1.4 | 0.072 ± 0.002 | 0.131 ± 0.005 | 0.150 ± 0.004 |
| ta-C(51) | 3.9 ± 1.3 | 9.0 ± 2.0 | 9.8 ± 1.5 | 0.025 ± 0.002 | 0.054 ± 0.028 | 0.060 ± 0.020 |
| ta-C(66) | 3.8 ± 0.6 | 9.1 ± 0.4 | 9.9 ± 0.4 | 0.044 ± 0.010 | 0.078 ± 0.009 | 0.089 ± 0.008 |
| ta-C(78) | 10.9 ± 0.8 | 10.4 ± 4.3 | 15.3 ± 2.5 | 0.099 ± 0.013 | 0.080 ± 0.015 | 0.128 ± 0.012 |

The densities $\rho$ and percentages of $sp^3$ carbon atoms $p_{sp^3}$ are estimated from Young's modulus using empirical formulae suggested by Ferrari and co-workers[45,46]. The reported RMS heights $h_{rms}$ and slopes $h'_{rms}$ are average values measured inside the wear tracks of DLC-coated cylinders and discs after running-in (800 cycles for ta-C(78) and 2000 cycles for the others). The mathematical definitions of the RMS heights and slopes are available in Supplementary Note 1. The composite RMS slopes are defined as $\sqrt{(h'_{rms,cylinder})^2 + (h'_{rms,disc})^2}$. The numbers after the ± signs are standard deviations.

and ta-C(78) (Fig. 1d). The rubbed surfaces are much rougher than the pristine surfaces (interferometer images in Supplementary Fig. 2). The average wear scar depths are about 0.5 and 1.5 μm for the disc and cylinder, respectively. Conversely, the wear scars of the softer DLCs are barely visible under the same observation conditions. Thus, ZDDP has a negligible effect on friction and wear of soft/moderately-hard DLCs but a catastrophic effect on those of harder ta-Cs.

**Contact mechanics calculations.** Our friction experiments are carried out with the same initial macroscopic cylindrical Hertzian contact pressure ($P_{Hertz} = 210$ MPa) for all tribo-pairs. However, the DLC coatings differ in their mechanical properties and surface topographies (Table 1) and thus local contact pressures $P_1$ at the asperity level can be different. Here, we perform contact mechanics calculations using the numerical boundary element method with experimental surface topographies to estimate these effects at the nanoscale (details in Methods). Surface topographies of all five DLC coatings are measured by atomic force microscopy (AFM) with a scan area of $20 \times 20$ μm$^2$ inside the wear tracks of cylinders and discs after running-in. The AFM topographies of cylinder and disc are brought into contact under an external pressure of 210 MPa. For the softer DLCs (a-C:H, a-C, and ta-C (51)), the $P_1$ probability density distributions have a high peak around $P_1 = 0$ and decay rapidly, with most asperities experiencing a $P_1$ lower than a few GPa (Fig. 1e). In contrast, the $P_1$ distributions for ta-C(66) and ta-C(78) are broader and contact pressures $P_1 > 10$ GPa occur frequently.

Figure 1f shows the effective local contact pressure $P_{1,eff}$ (as the median of the $P_1$ distribution over all contact spots where $P_1 > 0$) as a function of a material parameter $E^* h'_{rms}$ (the product of the contact modulus $E^*$ and the RMS slope $h'_{rms}$, discussed later on). For ta-C(66) and ta-C(78), $P_{1,eff}$ is 9.2 and 10.5 GPa, respectively. In contrast, for the three softer DLCs, $P_{1,eff}$ values are in the range 1.7–3.5 GPa. Interestingly, $P_{1,eff}$ for ta-C(51) is smaller than $P_{1,eff}$ for a-C (which has a larger $h'_{rms}$). The monotonic increase in $P_{1,eff}$ with $E^* h'_{rms}$ indicates that $P_{1,eff}$ is not only a function of the Young's modulus $E$ but also depends on surface topography.

**Surface chemical analyses.** Surface chemical analyses are performed using X-ray photoelectron spectroscopy (XPS) inside and outside the wear tracks of the discs to study the mechanisms underlying the tribological behaviours of DLC surfaces under PAO + ZDDP lubrication. The binding energies (BEs) of the C1s, O1s, P2p, S2p and Zn2p$_{3/2}$ photo-peaks are analysed carefully (see Methods for details of the fitting procedure and XPS spectrometer). Figure 2a shows the relative chemical compositions estimated from XPS spectra in the topmost 10 nm of the DLC discs. The ZDDP-derived elements, namely P, S, and Zn, are detected inside the wear tracks of all DLC surfaces (Fig. 2a), whereas very small traces of these elements are found outside the wear tracks. The atomic concentrations of P, S, and Zn inside the wear tracks decrease in the order: a-C:H>a-C>ta-Cs.

While the chemical states of C, O, P, and Zn on the five DLC coatings (Supplementary Fig. 3) are hardly distinguishable, this is not the case for sulphur. Figure 2b shows S2p XPS spectra inside and outside wear tracks on flat specimens. Outside the wear tracks, similar S2p XPS spectra are observed for all DLC surfaces with a peak at BE = 162.4 eV (Fig. 2b, right column), which is very close to that of sulphur in neat ZDDP (BE = 162.5 eV). The other peak at BE = 168.5 eV indicates the presence of sulphates (presumably due to thermal oxidation of DLC surfaces and subsequent chemisorption of ZDDP). In contrast, inside the wear track of a-C:H (Fig. 2b, left column), the S2p spectrum has one single component (in blue) at BE = 162.3 eV corresponding to S−Zn[7,24], or S−P and S=P in thiophosphates[24]. This peak is present also inside the wear tracks of the other DLCs, but the spectra contain another component (in red) at BE = 163.3 eV due to S−C bonds[25]. Note that C−S−H, C−S−C, or C−S−S−C bonding states are practically indistinguishable as their differences in the BEs are only about 0.1 eV[25]. The other component for ta-Cs is attributed to a peak at BE = 167.9 eV due to sulphur oxides[26,27]. Interestingly, the contribution from the S−C component at BE = 163.3 eV is much smaller than the one at BE = 162.3 eV for the a-C and ta-C(51), and thus sulphur in these coatings is preferentially bound to zinc and phosphorous (in the form of S−P and S=P). In contrast, for ta-C(66) and ta-C(78), the BE = 163.3 eV contribution is dominant, indicating the breaking of S−Zn bonds in ZDDP and the formation of S−C bonds. These results indicate that ZDDP is likely to stay intact on the three softer

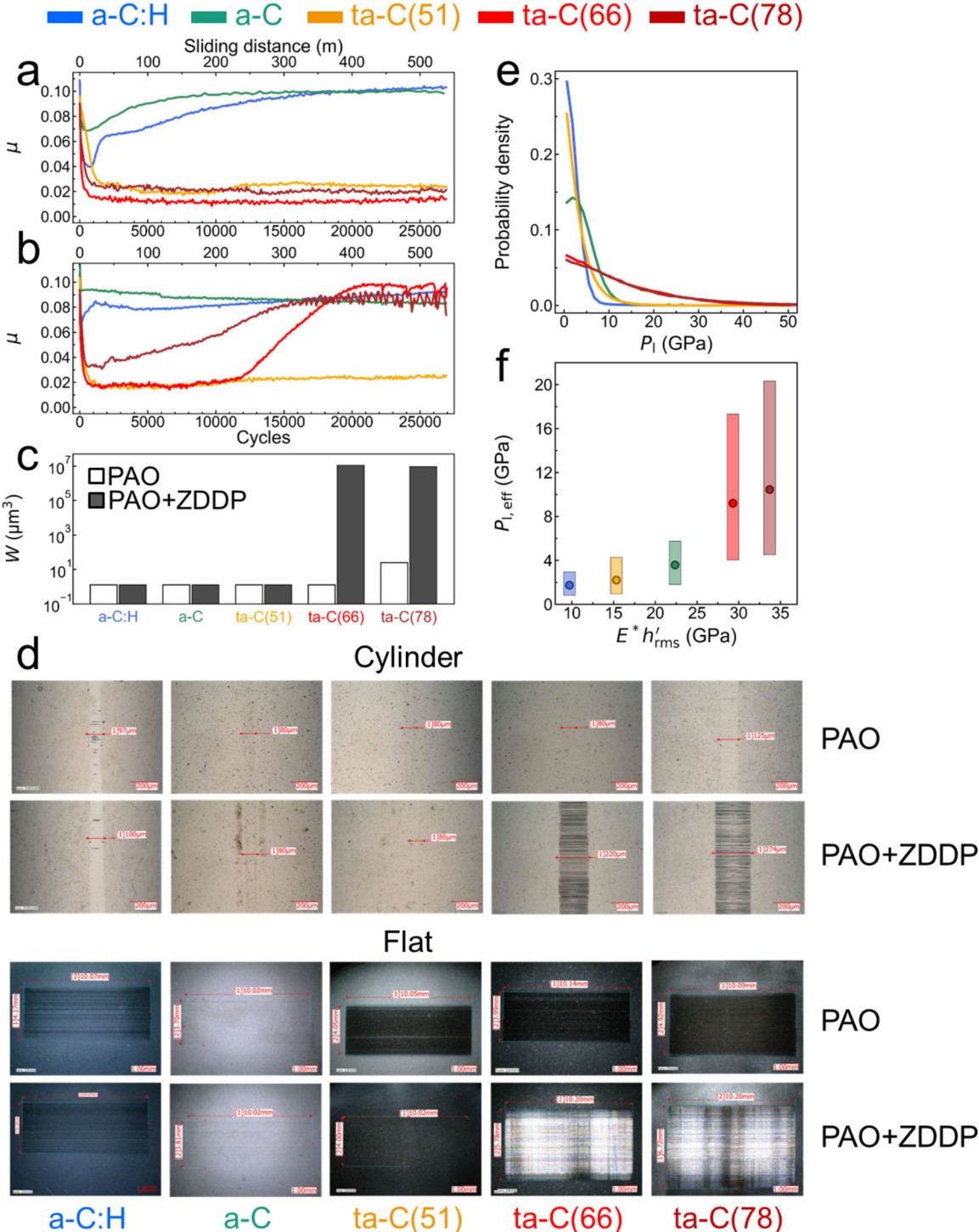

**Fig. 1 Reciprocating sliding tests of five self-mated DLC tribopairs.** Friction curves in pure PAO-4 base oil (**a**) and a mixture of PAO-4 and 1 wt% ZDDP (**b**) for a-C:H (blue), a-C (green), ta-C(51) (orange), ta-C(66) (red) and ta-C(78) (brown). **c** Composite wear volumes measured on the discs and cylinders by optical interferometry and **d** optical images of the wear tracks of the DLC-coated cylinders and discs after sliding. **e** Local contact pressure distributions calculated from contact mechanics calculations using AFM topographies measured inside the wear track after running-in (2000 cycles) with a scan area of $20 \times 20 \ \mu m^2$. **f** Effective local contact pressure $P_{l,eff}$ as a function of $E^*h'_{rms}$, where $E^*$ is the contact modulus and $h'_{rms}$ is the combined root-mean-square slope between two DLC surfaces. $P_{l,eff}$ is defined as the median over all nonzero $P_l$ values and is obtained from a numerical boundary element solution of two experimental DLC topographies under a normal pressure of 210 MPa. Filled circles (with the same colour coding as in **a** and **b** mark the material parameter $E^*h'_{rms}$ of each DLC coating. The boxes represent the range of the $P_l$ distribution (limited by the 25th and 75th percentiles).

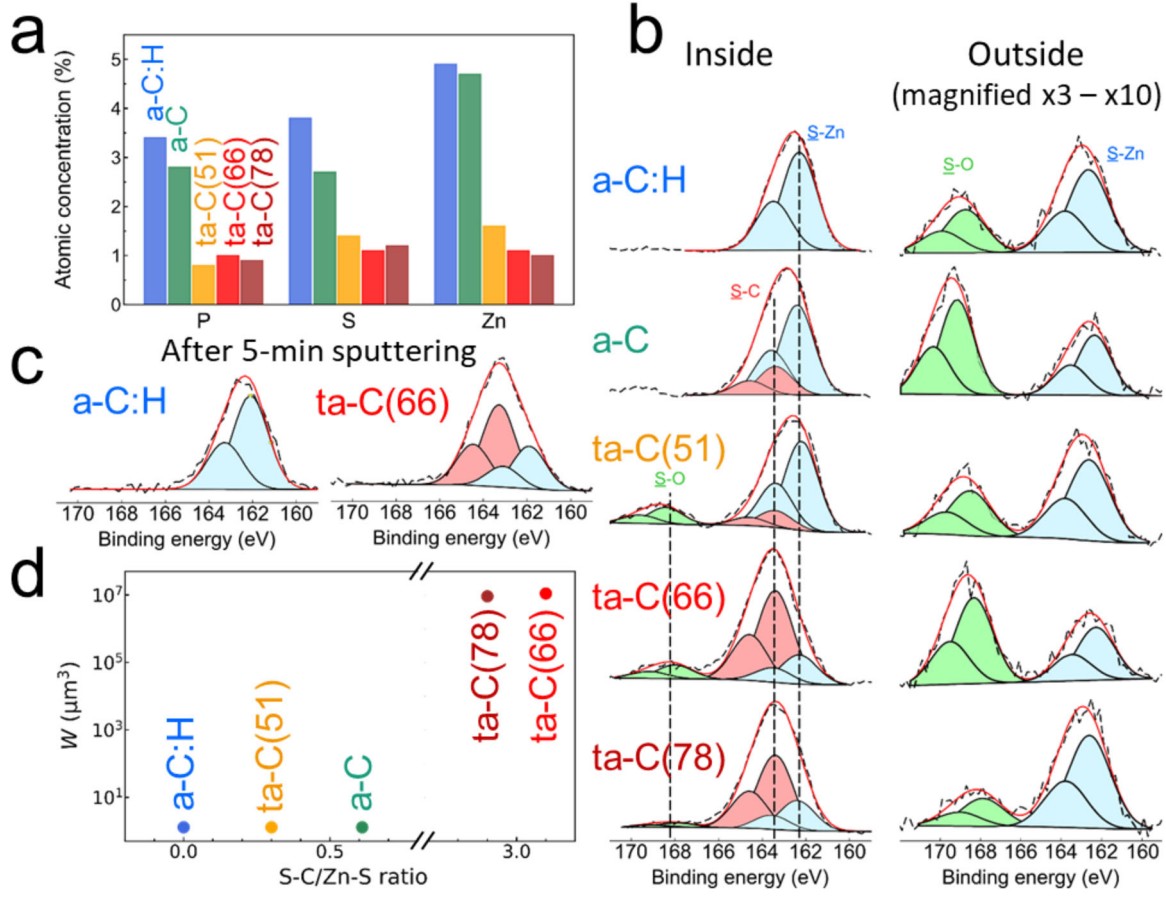

**Fig. 2 XPS surface chemical analyses for the five self-mated DLC tribopairs inside and outside wear tracks after sliding in PAO+ZDDP. a** Relative chemical compositions of ZDDP-derived chemical elements P, S and Zn inside the wear tracks. **b** S2*p* XPS spectra inside (left) and outside (right) the wear tracks. They are fitted using a doublet due to spin-coupling with a separation of 1.2 eV and an intensity ratio of 0.5. Peak coloured in blue correspond to S−Zn or S−P bonds, while red and green peaks correspond to S−C and S−O bonds, respectively. The S2*p* XPS spectra outside the wear tracks are magnified by a factor of 3 (for the ta-Cs) and a factor 10 (for a-C:H and a-C) to improve visibility. **c** S2*p* XPS spectra inside the wear track for a-C:H (left) and ta-C(66) (right) after 5-min sputtering (corresponding to a sputter depth of about 0.5 nm). **d** Wear volume *W* as a function of the S−C/S−Zn ratio. For a critical value of the S−C/S−Zn ratio (located somewhere between 1 and 2.5) a transition from negligible to high wear can be observed.

DLCs, whereas it undergoes decomposition on the hard ta-Cs. Moreover, for ta-C(66), the XPS spectra after 5-minute sputtering (corresponding to a sputter depth of about 0.5 nm[28]) remain similar to those before sputtering inside the wear track (Fig. 2b) and show the presence of significant amounts of S−C bonds (see Fig. 2c). In contrast, for a-C:H, no peaks attributed to S−C bonds are observed. This indicates that sulphur is present in the subsurface region of hard ta-Cs but not of a-C:H. Figure 2d shows a clear correlation between wear volume and S−C/S−Zn ratio.

**Scanning and transmission electron microscopy characterisations.** It is still not clear whether ZDDP-derived tribofilms form on DLC surfaces. Thicknesses, morphologies and chemical compositions of ZDDP-derived tribofilms are scrutinised using Scanning Electron Microscopy (SEM), Energy Dispersive X-Ray Spectroscopy (EDX) and Transmission Electron Microscopy (TEM). Figure 3 shows representative SEM images of the five DLC surfaces after sliding in PAO + ZDDP. ZDDP-derived tribofilms form on a-C:H, a-C, and ta-C(51) (Fig. 3a–c), as evidenced by superimposing the chemical map of zinc. Quantification results of EDX spectra are available in Supplementary Fig. 4 and Supplementary Table 2. On a-C:H, ZDDP-derived tribopatches are heterogeneous in size, ranging from 0.1

to 1 μm (Fig. 3a). On a-C, tribopatches are more homogenous in size and distribution (Fig. 3b). On ta-C(51), tribopatches are found in the least amount (Fig. 3c), in agreement with the XPS results. In all these cases the tribo-patches form predominantly at the edges of the stroke (Supplementary Fig. 5). For ta-C(66) and ta-C(78), scratches are observed and no ZDDP tribopatches are visible (Fig. 3d, e). Instead, small ZDDP-derived particles (marked by red circles) are detected using punctual EDX spectra.

The chemical structure and thickness of the ZDDP-derived tribofilms on a-C:H are identified by inspecting a cross-section of the a-C:H coating that exhibits the largest tribopatches after sliding using Focused Ion Beam preparation and TEM analysis. A bright-field (BF) TEM image reveals that the ZDDP-derived tribolayer (marked by a dashed green ellipse) is amorphous and discontinuous (Fig. 4a). Its thickness is about 20 nm, i.e. 5−10 times thinner than typical pad-like tribofilms on steel surfaces under similar sliding conditions[14], and in agreement with a previous ball-on-disc rolling/sliding experimental result[7]. Figure 4b shows a high-angle annular dark-field (HAADF) image of the tribofilm and the chemical compositions in three different regions of interest (ROI, dashed boxes). The presence of O, S and Zn inside the tribofilm (ROI 1) is confirmed by elemental quantitative analyses and a combined EDX chemical mapping for Zn, Pt, and C (Fig. 4c and Supplementary Table 3). Note that

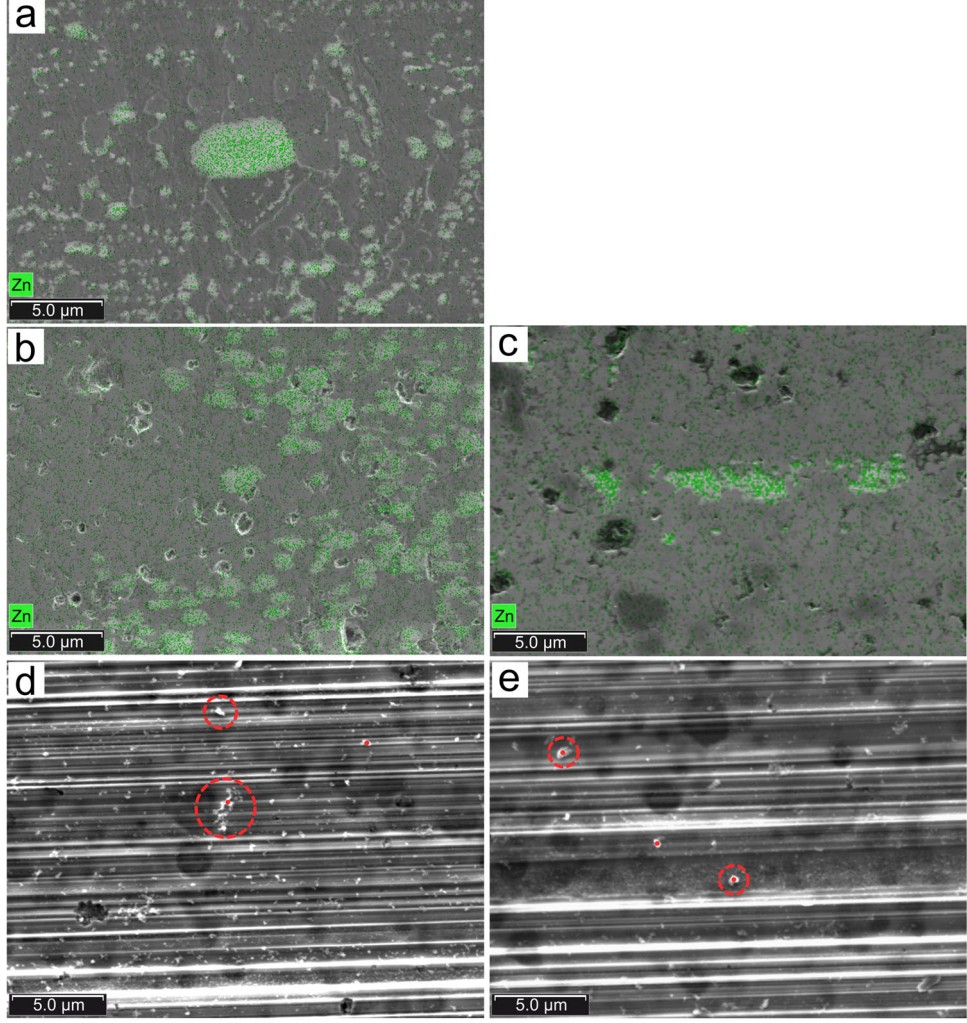

**Fig. 3 Representative SEM images of DLC surfaces after sliding in PAO+ZDDP.** SEM images for **a** a-C:H, **b** a-C, **c** ta-C(51), **d** ta-C(66), and **e** ta-C(78). ZDDP-derived species are indicated by green dots in **a−c** and by dashed red circles in **d** and **e**.

phosphorus is not resolvable since its signal overlaps with the platinum signal.

Combined BF-TEM images of the cross-section of a ta-C(66) sample after sliding show the formation of large grooves due to wear and filling of a groove with smeared wear particles (indicated by arrows in Fig. 4d). A clear boundary between the smeared layer of wear debris and ta-C bulk is observed (Fig. 4e). Chemical information is obtained by EDX inside the smeared layer and ta-C bulk (for ROI 4−8 specified in Fig. 4f). While ROI 4−6 contain sulphur and less zinc, ROI 7 and 8 contain mostly carbon and traces of oxygen and the chemical compositions are typical of the native ta-C. This indicates that sulphur is transported into the ta-C subsurface region. Interestingly, a higher magnification image (Fig. 4g) reveals that the sulphur-rich bright regions (ROI 4−6) undergo phase transformation and exhibit ordered graphitic structures, which is consistent with a previous study by Berman et al. showing sulphur diffusion into diamond nanoparticles and subsequent formation of an onion-like structure[29].

In summary, our friction experiments combined with surface analyses and contact-mechanics calculations suggest that friction and wear of DLCs in PAO + ZDDP depend on asperity-scale contact pressures. For softer DLCs, the majority of asperities experience mild local contact pressures (less than a few GPa), which seems to be sufficient for the growth of a 20-nm-thick, patchy tribofilm but insufficient for the complete decomposition of ZDDP. In contrast, for harder DLCs, higher $P_{l,eff}$ (>9 GPa) might promote breaking of Zn−S bonds in ZDDP and preferential formation of S−C bonds, accompanied by penetration of sulphur into the sub-surface region and massive wear.

**Quantum-chemical calculations.** The experimental correlation between wear and the presence of S−C bonds (Figs. 2 and 4) can be explained by weakening of ta-C via incorporation of small amounts of sulphur (Fig. 5). We carry out MD homogeneous shearing simulations of sulphur-doped DLCs using a third-order density-functional tight-binding (DFTB3) method[30]. Lees-Edwards boundary conditions[31] are employed in this non-equilibrium MD simulations to impose a shear flow in a representative volume element of a bulk a-C, a-C:H and ta-C system (see "Methods" for samples preparation).

For all three systems, the average shear stress $\tau$ (the yield stress) decreases with increasing sulphur concentration $C_S$ (Fig. 5b). While $\tau$ is almost identical for a-C and a-C:H, it is larger for ta-C (as expected). For a-C:H and a-C (Fig. 5b, red and blue curves, respectively), the yield stresses at $C_S = 0$ at% are about 11–12 GPa, i.e. almost the same as in ta-C at $C_S = 8$ at.% (Fig. 5b, green curve). Thus, significant sulphur doping of the hardest ta-Cs can lower their yield stress making the resulting sulphur-carbon phase much weaker than pure a-C:H and a-C.

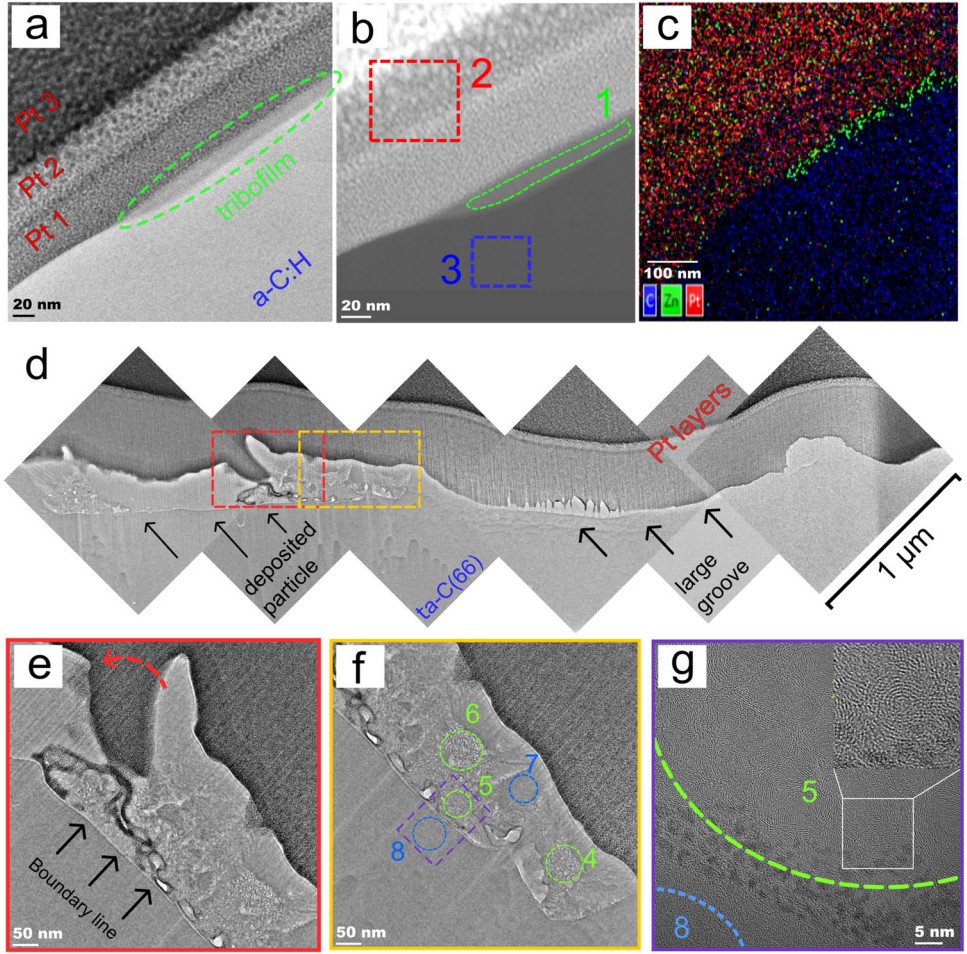

**Fig. 4 Cross-sectional images of DLC surfaces after sliding. a** BF TEM image of the cross section of a a-C:H surface. **b** HAADF image with recorded spectra from three different regions of interest (ROI). ROI 1: ZDDP-derived tribofilm, ROI 2: protective platinum layer, and ROI 3: a-C:H layer. **c** EDX map cartography superposition showing X-rays signal signatures of platinum (red), carbon (blue), and zinc (green). **d** Combined BF TEM images of the cross-section of a ta-C(66) surface. **e, f** TEM images of a boundary between the smeared layer of wear debris and ta-C bulk (indicated in **d** by the red and yellow box for **e** and **f**, respectively). In **f**, the ROIs used for EDX analysis are specified. **g** A high magnification TEM image of wear particles smeared on the ta-C (66) surface (indicated by the purple box in **f**).

But how does sulphur doping occur and how do reactions of ZDDP depend on coating properties such as stiffness, surface topography and chemical structure? Our experiments and contact mechanics calculations (Figs. 1–4) indicate that local contact pressures $P_l$ between surface asperities vary with the contact modulus $E^*$ and RMS slope $h'_{rms}$ of the coatings. Differences in both $P_l$ and surface reactivity of the DLC surfaces may trigger different mechanochemical reactions of ZDDP and thus influence the macroscopic tribological behaviour. Thus, additional simulations are performed in order to elucidate the dependence of ZDDP decomposition reactions on contact pressure and DLC chemical structure. DLC/DLC tribocontacts in presence of ZDDP are modelled using both quasi-static quantum-mechanical molecular statics (QMS) and QMD simulations (computational details in "Methods"). Two types of DLC surfaces are considered: (i) an a-C:H with bulk density $\rho = 2.0$ g cm$^{-3}$ and hydrogen concentration $C_H = 20$ at% and (ii) an a-C with the same density. The a-C sample represents the near-surface region of ta-C coatings, since ta-C is covered with a several-nm-thick a-C layer due to the nature of the deposition process[32] and a tribo-induced $sp^3$-to-$sp^2$ rehybridisation[33].

Figure 6a displays an initial configuration of the QMS system, where a ZDDP molecule with ethyl groups is positioned between

two a-C:H surfaces. Unpassivated surfaces are considered since asperity collisions can remove passivating species mechanically, leaving behind reactive carbon atoms[33]. A full hydrogen passivation would inhibit reactions of ZDDP even under extremely high contact pressures ($P_z > 10$ GPa) (Supplementary Fig. 6). As the two surfaces come gradually into contact, ZDDP reacts with the upper surface even at a low contact pressure ($P_z \approx 0.1$ GPa in Fig. 6b). The reactions start with the breaking of a Zn−S bond in the molecule and the formation of a S−C bond between the molecule and the upper surface (Fig. 6c). The second chemical bond forms between the Zn atom in the molecule and a C atom of the lower surface, resulting in the formation of a cross-linked configuration that bridges the tribogap at a slightly larger $P_z \approx 0.3$ GPa (Fig. 6d). At $P_z = 5.1$ GPa, breaking of another Zn−S bond and formation of another S−C bond are observed (Fig. 6e). As the contact is reopened, the cross-linked ZDDP molecule is stretched (corresponding to negative $P_z$; grey line in Fig. 6b). This mechanical pulling induces breaking of a S−C bond on the lower surface and reforming of a Zn−S bond in the molecule, and eventually fragmentation of the ZDDP molecule into two smaller products (Fig. 6f). It is important to note that reactions of ZDDP on surfaces differ from those observed during simulations in the gas phase, where Zn−S bonds repeatedly break

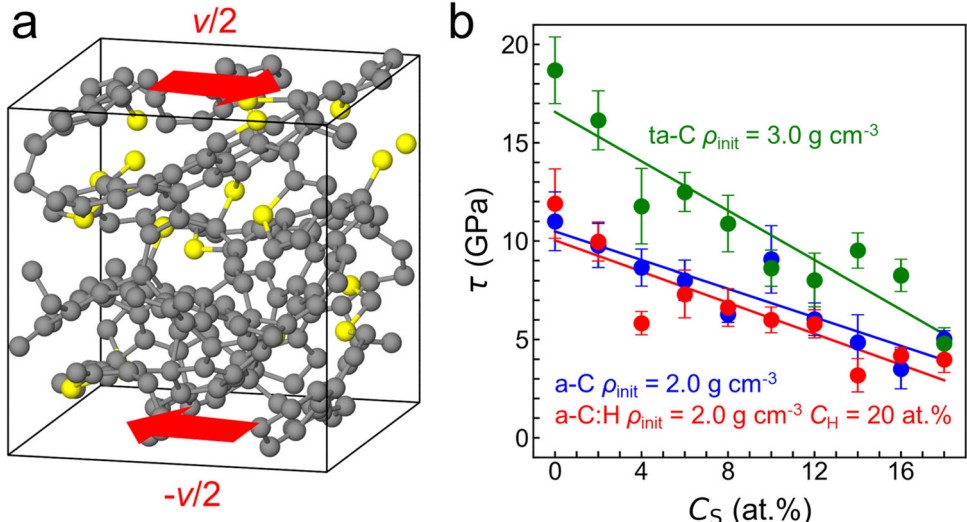

**Fig. 5 DFTB3 quantum MD simulations of sulphur doped DLCs subject to homogeneous shear deformation. a** Atomic configuration at $t = 50$ ps for a-C with an initial density of 2.0 g cm$^{-3}$ and 8 at% S. Carbon atoms are represented by grey and sulphur atoms by yellow spheres. A shear velocity of $v = 100$ m s$^{-1}$ is imposed along the $x$ direction (as shown by red arrows) with Lees-Edwards boundary conditions. **b** Shear stresses $\tau$ as a function of the sulphur concentration $C_S$ in ta-C with an initial density $\rho_{init} = 3.0$ g cm$^{-3}$, in a-C with $\rho_{init} = 2.0$ g cm$^{-3}$, and in a-C:H with $\rho_{init} = 2.0$ g cm$^{-3}$ and 20 at% hydrogen. $\tau$ is calculated every 10 ps and averaged for 50-ps MD runs. Error bars represent standard error of means.

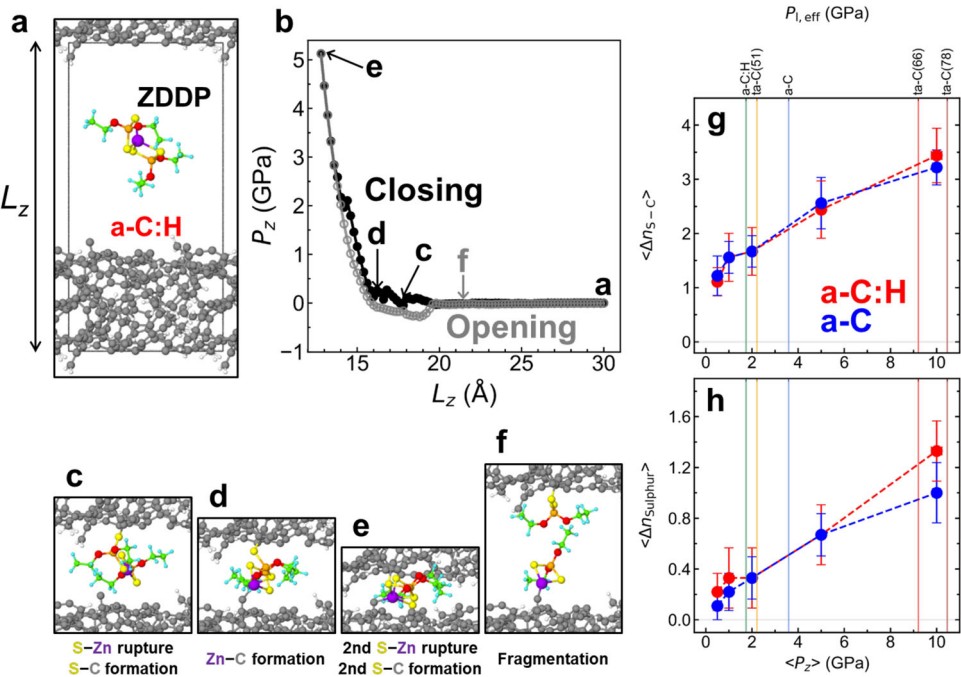

**Fig. 6 Quasi-static contact-closing/opening simulations for a-C:H with $\rho = 2.0$ g cm$^{-3}$ and CH = 20 at% and a-C with $\rho = 2.0$ g cm$^{-3}$ in contact with a ZDDP molecule using DFTB3 quantum chemical calculations. a** Initial atomic configuration for a representative a-C:H system. **b** Evolution of the normal pressure $P_z$ as a function of the $z$ system dimension $L_z$ during the closing (black circles) and opening process (grey circles). **c-f** Evolution of atomic configurations for a representative trajectory of a-C:H. For C and H, different colours are assigned between the surface atoms (grey for C and white for H) and atoms in the ZDDP molecule (green for C and cyan for H). Zn, S, P and O are represented by violet, yellow, orange, and red spheres, respectively. The $L_z$ values and corresponding normal pressures of the snapshots **c-f** are indicated by labels in **b**. **g** Averaged numbers of S−C bonds $\langle \Delta n_{S-C} \rangle$ formed between ZDDP sulphur and DLC carbon atoms as well as **h** sulphur atoms released from ZDDP $\langle n_{Sulphur} \rangle$ to the DLC after reopening the contact as a function of the contact pressure $P_z$ for a-C:H with $\rho = 2.0$ g cm$^{-3}$ and $C_H = 20$ at.% (red) and a-C with $\rho = 2.0$ g cm$^{-3}$ (blue). The error bars represent standard error of the means. The vertical lines mark the effective local contact pressures $P_{l,eff}$ for the five experimental coatings (Fig. 1f).

and form and ZDDP's decomposition starts with the loss of alkyl groups[34]. On surfaces, breaking of Zn−S bonds is the first observed reaction because the formation of S−C bonds with C atoms on the surface prevents the reformation of the Zn−S bonds.

Bond-breaking and -forming processes in ZDDP are strongly affected by the contact pressure but not by the chemical structure of the DLC surfaces. Figure 6g shows the contact-pressure dependence of chemical reactions of ZDDP squeezed quasi-statically between two a-C and two a-C:H surfaces. The values

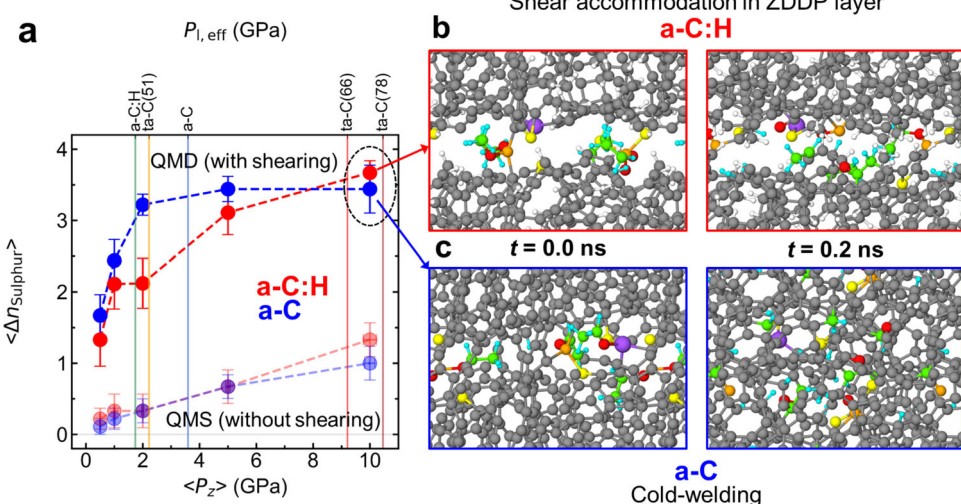

**Fig. 7 DFTB3-MD sliding simulations of a-C:H and a-C in contact with a ZDDP molecule.** A shear velocity of 30 m s⁻¹ is imposed along the $x$ direction using Lees-Edwards boundary conditions. The system temperature is kept constant at $T = 300$ K. **a** Averaged numbers of sulphur atoms $\langle n_{Sulphur} \rangle$ released from ZDDP to DLC after reopening the contact as a function of the contact pressure $P_z$. The error bars represent standard error of the means. The vertical lines mark the effective local contact pressures $P_{l,eff}$ for the five experimental coatings (Fig. 1f). As references, $\langle n_{Sulphur} \rangle$ for quasi-static calculations with no shear (Fig. 6h) are depicted with semi-transparent markers. Representative snapshots of DLC surfaces at $P_z=10$ GPa for **b** a-C:H and **c** a-C. The left and right pictures in **b** and **c** show configurations at $t = 0.0$ and 0.2 ns, respectively.

$\langle n_{S-C} \rangle$ on the vertical axis represent the average number of bonds between sulphur and DLC carbon atoms that remain intact after reopening the contacts. Our simulations reveal that the formation of S−C bonds is the most dominant chemical reaction, in agreement with our XPS results (Fig. 2). While $\langle n_{S-C} \rangle$ increases with $P_z$ ($0.5 \le P_z \le 10$ GPa), the pressure dependence of $\langle n_{S-C} \rangle$ in a-C:H and a-C is almost identical. For both DLC surfaces, chemical reactions of ZDDP follow the scenario described in Fig. 6. At $P_z \lesssim 2$ GPa, most ZDDP molecules chemisorb without decomposition (Fig. 6c) or decompose into two fragments by breaking of Zn−S bonds. A further increase of the contact pressure induces decomposition of ZDDP into atoms and small fragments.

Only sulphur atoms that are effectively released onto the DLC surface are relevant to the weakening of the DLC presented in Fig. 5. Therefore, we next calculate the average number of sulphur atoms $\langle n_{Sulphur} \rangle$ that are not bound to any atoms in ZDDP and remain chemisorbed on the DLC surface after reopening the contact (Fig. 6h). As already observed for $\langle n_{S-C} \rangle$, $\langle n_{Sulphur} \rangle$ increases significantly with $P_z$ and no difference between a-C:H and a-C is detectable (Fig. 6h). Even though the a-C systems used here are representative of the surface region of ta-C, which typically has a low $sp^3$ content of about 10%[33], we performed a comparison between ta-C with high $sp^3$ percentages $p_{sp^3}$ (ranging from 48−67%) and a-C with $p_{sp^3} = 2 - 9\%$ and observed that $\langle n_{S-C} \rangle$ and $\langle n_{Sulphur} \rangle$ are not significantly affected by the $sp^3$ content of the hydrogen-free DLC surface (Supplementary Fig. 7).

Next, we perform QMD sliding simulations to explore the role of shearing (Fig. 7). A DLC/ZDDP/DLC triobinterface is studied at a sliding velocity of 30 m s⁻¹ and various $P_z$ values (see snapshots for $P_z = 10$ GPa in Fig. 7b, c). After 0.2 ns sliding, the two DLC surfaces are separated quasi-statically to ensure the consistency with the QMS results. For both a-C and a-C:H, $\langle n_{Sulphur} \rangle$ increases about 2−3 times at all $P_z$ compared with $\langle n_{Sulphur} \rangle$ in QMS simulations and the effect of the DLC's chemical structure on the effective number of sulphur atoms

left on the surface is not significant (except for $P_z = 2$ GPa) as shown in Fig. 7a.

Thus, the chemical difference between a-C:H and ta-C, i.e. presence or absence of hydrogen, hardly affects the kinetics of ZDDP's decomposition. The local contact pressures alone determine sulphur's chemical state on the different DLC surfaces. However, the DLC's chemical structure affects shear accommodation at the DLC/ZDDP/DLC interface at high contact pressures. Similar surfaces functionalized with ZDDP-derived fragments form at small contact pressures ($P_z \lesssim 5$ GPa) on a-C:H and a-C (Supplementary Fig. 8), whereas a clear structural difference is observed at $P_z = 10$ GPa (Fig. 7b and c). While a layer of ZDDP-derived fragments still accommodates shear on a-C:H, the a-C surfaces are cold-welded (Fig. 7c and Supplementary Movies 1 and 2). In this case, ZDDP-derived atoms/fragments diffuse underneath the surface as a result of mechanical mixing.

Figure 8 provides further statistical evidence that the chemical difference between the DLC surfaces (i.e. the hydrogen concentration) has a significant effect on the shear response of the frictional interface. First, the weakest interface in each final configuration after MD sliding is determined and then the number of cold-welding C−C bonds $n_{C-C}^{CW}$ within this interface is calculated. Averaging over different MD runs yields the mean density of cold-welding C−C bonds $\langle n_{C-C}^{CW} \rangle$ for a-C:H and a-C (Fig. 8a). $\langle n_{C-C}^{CW} \rangle$ is almost zero at $P_z \lesssim 2$ GPa for both a-C:H and a-C, indicating the presence of a sliding interface with no significant cold welding. However, $\langle n_{C-C}^{CW} \rangle$ increases drastically for a-C at $P_z \gtrsim 5$ GPa, which is a typical contact pressure range reached only by ta-C(66) and ta-C(78) in our experiments (vertical lines in Fig. 8a). In this case, cold welding and shear-induced plastic flow increase the atoms' mobility and induce chemical mixing of the contaminated a-C surfaces. Figure 8b and c clearly shows that the depth distribution (z-axis profile) of the sulphur density at $P_z = 10$ GPa is significantly broadened for a-C, while sulphur is still localised at the surfaces for a-C:H.

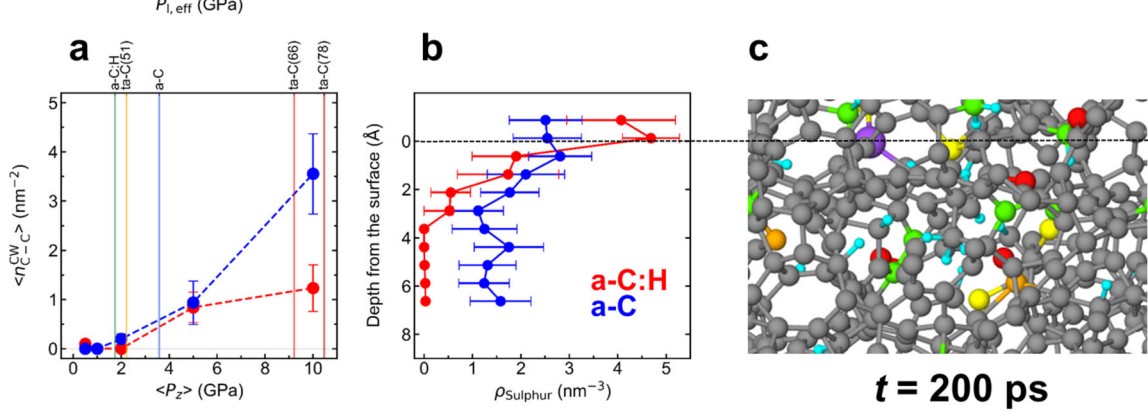

**Fig. 8 Different chemical responses to shear between a-C:H and a-C. a** Mean bond densities of cold-welding C−C bonds $\langle n_{C-C}^{CW} \rangle$ at $t = 0.2$ ns. The error bars represent standard error of the means. $n_{C-C}^{CW}(z)$ is defined as the number of C−C bonds across a $x$–$y$ plane at height $z$. For each system, the minimum values $n_{C-C}^{CW} = \min_z n_{C-C}^{CW}(z)$ (corresponding to a shear interface) is searched with a step of $\Delta z = 0.5$ Å. **b** Depth profile of the number density of sulphur $\rho_{Sulphur}$ (nm$^{-3}$) averaged over the last 0.1 ns MD runs at $P_z = 10$ GPa. The error bars represent standard error of the means. **c** Example of a chemically mixed a-C layer at $t = 0.2$ ns. The horizontal dashed line represents an initial a-C surface before sliding.

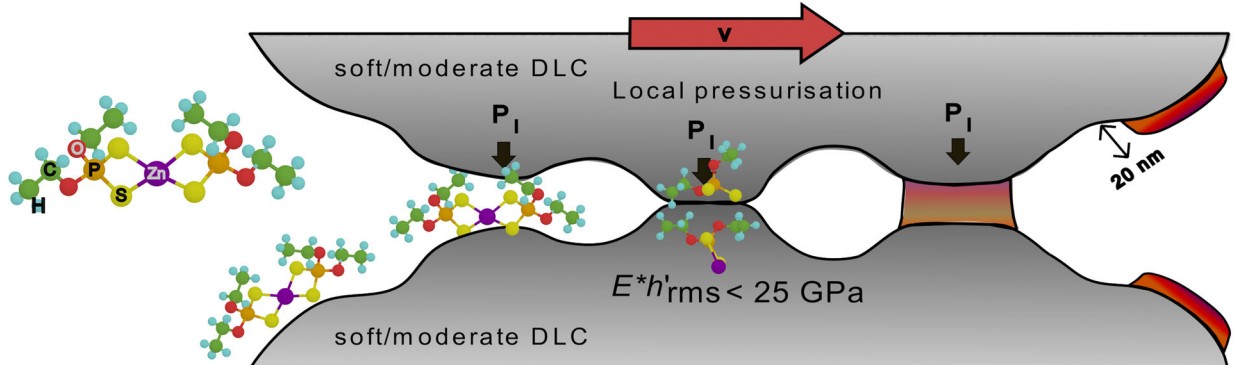

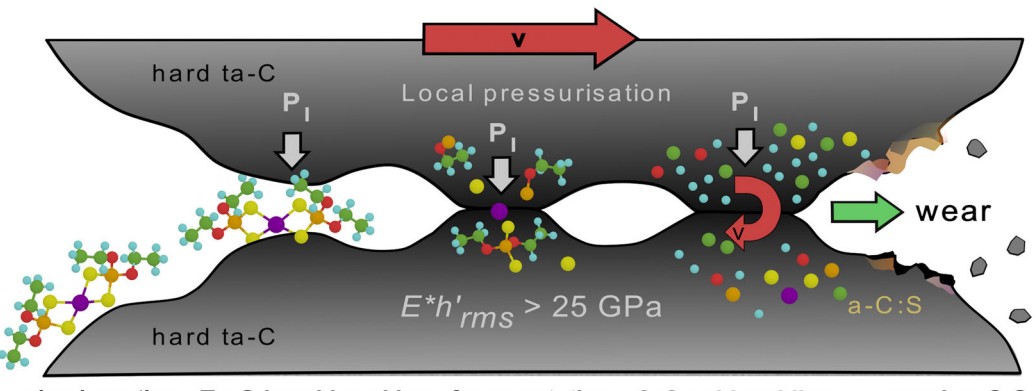

**Fig. 9 Schematic representation of likely scenarios for DLC coatings lubricated by ZDDP-additivated oils. a** Formation of ZDDP-derived patchy anti-wear tribofilms on soft DLC coatings (when $E^* h'_{rms} < 25$ GPa) and **b** wear of hard ta-C coatings.

## Discussion

In this study, a combination of experiments and multiscale modelling reveals the crucial impact of both contact mechanics of surface asperities and DLC chemical structure on the tribochemistry of ZDDP decomposition and, in turn, on the different macroscopic tribological behaviours of DLC coatings in boundary lubrication with a ZDDP-additivated base oil (see a schematic representation of our findings in Fig. 9). While the macroscopic cylindrical Hertzian contact pressure is identical in all our experiments ($P_{macro} = 210$ MPa), the nanoscale mechanical response of surface asperities to the normal load varies with contact modulus $E^*$ and RMS slope $h'_{rms} = \langle |\nabla h|^2 \rangle$ of the

coatings. According to Persson's contact mechanics theory[35,36], the average local contact pressure for small $P_{macro}$ is given by $\langle P_1 \rangle = \frac{h'_{rms}E^*}{\kappa}$ with contact modulus $E^* = \frac{1}{2}\frac{E}{1-\nu^2}$ and combined RMS slope of the two surfaces $h'_{rms} = \sqrt{\left(h'_{rms,cylinder}\right)^2 + \left(h'_{rms,disc}\right)^2}$. When the pressure distribution follows a Rayleigh formula[36], $\kappa$ is equal to $\sqrt{\frac{8}{\pi}} \approx 1.6$ and the median $P_{1,eff}^{Persson}$ can be also analytically calculated by $\sqrt{\frac{\ln 2}{2}}h'_{rms}E^* \approx 0.59 h'_{rms}E^*$ (irrespective of $P_{macro}$). A linear fitting of $P_{1,eff}$ obtained from our contact mechanics calculations gives a slope of 0.4 that is lower than the theoretical one. While this can be probably attributed to the lower degree of self-affinity of our DLC surfaces with respect to the theory and to overestimation of $h'_{rms}$ due to the presence of holes (Supplementary Fig. 1), Persson's theory provides the rationale behind the results of our contact mechanics calculations. Interestingly, the ordering in $P_{1,eff}$ between a-C and ta-C(51) shows that $P_{1,eff}$ depends on both contact stiffness and surface topography. While ta-C(51) is much stiffer than a-C, $h'_{rms}$ for ta-C(51) is almost half of that for a-C. This smaller $h'_{rms}$ for ta-C(51) can decrease $P_{1,eff}$ compared with a-C, which is in excellent agreement with the experimental S−C/S−Zn ratios (Fig. 2d).

Our results suggest that the effective local contact pressure $P_{1,eff}$ is a crucial parameter to determine the kinetics of ZDDP decomposition as well as the wear and friction regime. For soft DLCs (such as a-C:H, a-C and ta-C(51)), at $P_{1,eff} \lesssim 3$ GPa, ZDDP chemisorbs on the surfaces by breaking of Zn−S and formation of Zn−C bonds and undergoes decomposition into zinc- and dithiophosphates (Fig. 9a). An anti-wear tribofilm, consisting of zinc polyphosphates, can grow via accumulation and subsequent cross-linking of such precursors[21]. During the tribofilm growth, the formation of a narrow shear zone in ZDDP-derived layers inhibits DLC's weakening by mechanical mixing. Eventually, nanoscale patchy anti-wear tribofilms form on the surfaces. Conversely, $P_{1,eff}$ exceeds 9 GPa on hard ta-Cs and such high local contact pressures aggressively promote ZDDP fragmentation (Fig. 9b). Mechanochemical sulphur doping of the sub-surface region becomes prominent and weakens the carbon matrix via shear-induced mixing, resulting in massive wear of ta-Cs. The difference in wear volume between a-C:H and hard ta-Cs is consistent with previous experiments[5]. Interestingly, $P_{1,eff}$ values (Fig. 1f) are consistent with contact pressures for single-asperity DLC/steel sliding contacts where tribofilm growth was observed[16]. Since $P_{1,eff}$ is estimated as the median of all the local contact pressures in the real contact area from contact mechanics calculations, and thus characteristic of asperity-asperity contacts, this consistency suggests that the use of $P_{1,eff}$ can fill the gap between atomic and macroscopic experimental scales. $P_{1,eff}$ is determined by the contact parameter $E^* h'_{rms}$. Therefore, a direct link between coatings' properties and ZDDP's tribochemistry, which in turn governs the DLCs' macroscopic tribological behaviour, can be established. When $E^* h'_{rms}$ exceeds a critical value (~25 GPa), tribofilm formation is replaced by sulphur doping (Fig. 9).

In addition to contact pressure, shear stress plays a decisive role in lowering the activation barriers for ZDDP chemical reactions and increasing the accessibility of reactive sites. Moreover, shear action at high contact pressures ($P_1 > 5$ GPa) makes the effect of the DLC's chemical structure (i.e. presence or absence of hydrogen) apparent. At low contact pressures ($P_1 < 5$ GPa), irrespective of presence or absence of hydrogen, shear can be accommodated in a narrow region consisting of ZDDP-derived species. Thus, since $P_{1,eff} \lesssim 3.5$ GPa for a-C:H, a-C and ta-C(51),

these soft DLC surfaces are likely to remain intact, and the accumulation of ZDDP-derived species can lead to the growth of anti-wear tribofilms irrespective of the chemical differences between the DLC coatings.

However, at $P_1 > 5$ GPa, for a-C and ta-Cs, the separation between two surfaces is insufficient to localise shear in the ZDDP-derived tribolayer, and cold welding via C−C bonds (bridging the two surfaces) takes place. Under shear, mechanical mixing at the cold-welded tribointerface induces sulphur doping into the sub-surface regions resulting in significant DLC's weakening and wear. This process is mechanically driven and leads to a non-equilibrium state with low shear resistance that accommodates the shear deformation. Like in other mechanically driven tribological processes[37,38], this metastable material has a higher energy than the original one and sulphur penetration cannot be described thermodynamically. In contrast, on a-C:H, the ZDDP chemisorbed layer would be able to support $P_1 > 5$ GPa owing to steric hindrance between surface hydrogen atoms. Although such high local pressures are not reached with our a-C:H, they would be possible for a hard ta-C:H.

In conclusion, this study sheds light on hitherto unknown multiscale physical processes and reveals how the interplay of mechanical, topographical and chemical factors determine the tribological behaviour of DLC coatings under lubrication with a ZDDP-additivated base oil. We find a knowledge-based design recipe for DLC coatings (hydrogen-free and of moderate hardness) that are wear-resistant in ZDDP-additivated oils while still allowing ultralow friction. Furthermore, the insights into ZDDP's fundamental mechanochemistry and the interplay of mechanics and chemistry could be transferable to ferrous or other non-ferrous/metallic lubricated systems. Hopefully, this study will pave the way for further studies aiming at finding FMs and lubrication configurations/conditions that improve the wear and friction performance of DLC/ZDDP systems and, more in general, at controlling tribochemistry of additives by tuning microscopic contact mechanics and surface chemical structure. Such studies could facilitate the formulation of alternative, less harmful anti-wear additives that protect iron-based and DLC surfaces simultaneously.

## Methods

**Materials.** Five types of DLC coatings, labelled by a-C:H, a-C, ta-C(51), ta-C(66) and ta-C(78), were prepared by the company HEF IREIS. The numbers in the parentheses represent the hardness values of ta-C in GPa. Chemical compositions, mechanical and structural properties, and roughness parameters of the DLC coatings are tabulated in Table 1. The coatings had a thickness of 2 μm and were deposited onto mirror-polished steel discs and steel cylinders. The steel discs had a diameter of 25 mm and a thickness of 7 mm, and was made of M2 steel (0.85%C, 6%W, 5%Mo, 4%Cr, 2%V, hardness 64 HRC). The 100Cr6 steel cylinders had a radius of 5 mm and a length of 10 mm. For both geometries, an intermediate layer was applied onto the steel surface to improve adhesion of DLC films. In order to obtain a smooth surface and remove surface defects, the DLC-coated discs and cylinders were polished prior to tribological tests. The surface roughness was measured with a tactile profilometer using a filter cut-off of $\lambda_c = 80$ μm. The mechanical properties of the materials were determined by nanoindentation using a Nanoindenter XP®. Indents were done with a diamond Berkovich indenter (tetrahedral geometry, 115.12 ° between edges). The maximum applied load was 450 mN assuming a Poisson's ratio of 0.20. Continuous stiffness measurement method was used. An oscillating displacement with a frequency of 32 Hz and an amplitude of 1 nm was superimposed at the continuous displacement. The tests were performed using a "constant strain rate" procedure. The ratio of the loading rate to the load was kept constant and was equal to $3 \times 10^{-2}$ s$^{-1}$. With a Berkovich indenter, the strain is constant during loading experiments[39]. Seven tests are made for each sample. Simultaneous measurement of the applied normal load, the contact stiffness versus displacement was recorded. Thus, hardness $H$ (related to a contact pressure) and reduced elastic modulus $E^*$ were continuously calculated and analysed versus plastic depth. The mechanical properties of the DLC coatings on the cylinders are assumed to be the same as those on the discs. PAO-4 as a base oil and primary-C4 ZDDP (1 wt.%, sourced from Total, France) were used. We note that the concentration of 1 wt% is typical for commercial engine oils and that

primary C4 ZDDP is more thermally stable[40] and its tribofilm growth rate is slower than secondary ZDDP[15]. The kinematic viscosity of PAO-4 is 3.8 cSt at $T = 100\,°C$.

**Tribological tests**. Cylinder-on-disc sliding tests were performed using a home-made linear reciprocating tribometer[41]. All tests were carried out using self-mated DLC/DLC tribopairs. The contacting surface between the cylinder and disc (flat coupon) was immersed in 3 ml of lubricant. A good contact between the disc and cylinder is ensured for every test thanks to an auto-alignment spring system of the cylinder holder. The friction coefficient $\mu$ is calculated as the average of the instantaneous friction coefficient over one complete cycle (1000 points per cycle). The testing conditions used were as follows: applied load $F_N = 23$ N (corresponding to an initial maximum cylindrical Hertzian contact pressure $P_{Hertz}$ of 210 MPa), temperature $T = 110\,°C$, average speed $v = 0.1$ m s$^{-1}$, stroke length $l = 10$ mm, and test duration $t = 90$ min (27,000 cycles). $P_{Hertz} = 210$ MPa is the same for all DLC coatings. Since the DLC coatings are only 2 μm thick and much thinner than the steel substrate, the macroscopic contact pressure is dominated by mechanical properties of the steel substrate. Indeed, the diameter of the wear scars is equal to that predicted by the Hertz theory. In this test configuration, the lubrication regime corresponds to boundary lubrication (BL) at low speeds, and transitions into mixed lubrication (ML) at higher speeds, as the thickness of the lubricant film increases (see estimation of EHL film thickness and a detailed discussion about the lubrication regime in Supplementary Note 9). As shown in the following section, sliding induced local temperature rises in the BL sections of the stroke stay well below 5 °C.

**Calculation of flash temperatures**. In our reciprocating sliding tests, flash temperatures at the sliding interface do not affect ZDDP's decomposition and macroscopic friction and wear. For rectangular contacts (i.e. cylinder on flat)[42], the flash temperature rise ($T_f$) at a dimensionless position $X$ on a homogeneous body subject to a uniform heat flow $q = \frac{\mu W |v_{disk} - v_{cylinder}|}{4bl}$ is calculated by

$$T_f(X) = \frac{2q\kappa}{\pi K v} \int_{X-L}^{X+L} e^{-\eta} d\eta \int_0^B \frac{e^{-(\eta^2 + \zeta^2)^{\frac{1}{2}}}}{(\eta^2 + \zeta^2)^{\frac{1}{2}}} d\zeta, \qquad (1)$$

where $l$ and $b$ are the half width and length of the rectangular contact, respectively, $K$ is the thermal conductivity, μ is the friction coefficient, $W$ is the normal load, and $v$ is the sliding speed of the homogenous body relative to the contact. The thermal diffusivity $\kappa$ is calculated from $\kappa = K/\rho C_p$, where $\rho$ is the mass density and $C_p$ is the heat capacity. The dimensionless parameters $X$, $B$, and $L$ are defined as $\frac{vx}{2\kappa}, \frac{vb}{2\kappa}$ and $\frac{vl}{2\kappa}$. $x$ is the position on the body along the sliding direction. First, we consider the temperature rise of a homogeneous body consisting of steel and correct for the presence of a thin DLC coating in a following step. The material parameters for 100Cr6 steel $\rho_{steel} = 7.704$ g cm$^{-3}$, $K_{steel} = 26.1$ W m$^{-1}$ K$^{-1}$, and $C_{p,steel} = 0.446$ J g$^{-1}$ K$^{-1}$ are taken from the literature[43].

In the following, we choose a reference frame with a stationary cylinder in contact with a moving disk. The integral in the formula for $T_f(X)$ is evaluated numerically for the stationary situation ($v \to 0$) providing the maximum temperature rise $T_{fmax,cylinder}$ on the cylinder as well as for velocities $|v| > 0$ providing the maximum temperature rise $T_{fmax,disk}$ on the disk.

Now, we correct both flash temperatures for the presence of a coating. For a layered body consisting of a thin film (DLC) and substrate (steel), the maximum flash temperature for DLC $T_{fmax,DLC}$ is calculated from the ratio of $T_{fmax,DLC}$ to that for the homogeneous steel $T_{fmax,steel}$[44]:

$$\tau = \frac{T_{fmax,DLC}}{T_{fmax,steel}} = \frac{K_{steel}}{\alpha K_{DLC}} \left\{ 1 - \frac{2A}{1+A} f^*(A, Dp) \right\}, \qquad (2)$$

for $0.1 \le Pe_{DLC} \le 10.0$

$$f^*(A, Dp) = \frac{1}{1 + 2.329(1+A)^{-(0.475+0.88Dp)} Dp^{b(A)} \exp(-0.62Dp)}, \qquad (3)$$

$$b(A) = 0.206 - 0.0413A + (0.0406 + 0.022A - 0.0263A^2)^{0.5}, \qquad (4)$$

for $Pe_{DLC} < 0.1$,

$$f^*(A, Dp) = \frac{1}{\left(1 + \frac{3.36}{(1+A)^{1.097}} Dp\right)^{b(A)}}, \qquad (5)$$

$$b(A) = 1.42 + 0.399A - 0.116A^2 + 0.143A^3, \qquad (6)$$

where $Pe_{DLC} = \frac{vl}{2\kappa_{DLC}}$ and $Pe_{steel} = \frac{vl}{2\kappa_{steel}}$ are the Peclet numbers of DLC and steel, respectively, $A = \frac{\lambda - \alpha}{\lambda + \alpha}$, $\lambda = \frac{K_{steel}}{K_{DLC}}$, $\alpha = \sqrt{\frac{1+Pe_{DLC}}{1+Pe_{steel}}}$ for $0.1 \le Pe_{DLC} \le 10.0$ and $\alpha = 1$ for $Pe_{DLC} < 0.1$, and $Dp = D^2(1 + Pe_{DLC})$. The dimensionless thickness parameter $D$ is defined as $d/2l$, where $d$ is the thickness of a DLC coating ($d = 2$ μm).

According to articles by Robertson and his co-workers[45,46], the correlations between the density $\rho$ and Young's modulus $E$ of a DLC coating are given by

$$\rho = 1.37 + E^{2/3}/44.65 \text{ for a} - C \text{ and ta} - Cs, \qquad (7)$$

$$\rho = 0.257 + 0.011(E + 511)/44.65 \text{ for a} - C:H. \qquad (8)$$

The heat capacities $C_p$ are taken from the literature[47]. The thermal conductivities $K$ for the DLCs are estimated by a linear fitting of the relation between experimental values of $K$ and $E$[48]. A friction coefficient $\mu = 0.02$ is used for ta-C(51) and $\mu = 0.1$ for the other coatings. $T_{fmax}$ is defined as a maximum $T_f$ during the stroke.

At the end, the maximum flash temperature rise $T_{fmax}$ must be the same for both stationary cylinder and moving disk, and can be calculated from[49].

$$\frac{1}{T_{fmax}} = \frac{1}{T_{fmax,cylinder}} + \frac{1}{T_{fmax,disk}}. \qquad (9)$$

Most relevant are the flash temperature rise in the BL regime (i.e. at low sliding speeds near the edges of stroke) where reactions of ZDDP happen and thus our simulation models are valid. The detailed results are given in Supplementary Note 10. An average maximum flash temperature in the BL regime (18% of the stroke) $T_{fmax,BL}$ is 2.3 °C for a-C:H. For ta-C(51), the BL regime accounts for 32% of the stroke and $T_{fmax,BL}$ is 0.5 °C. The temperature rise is negligibly small for all DLCs, and indeed no correlations of $T_{fmax,BL}$ with DLC's properties and observed phenomena were found. For example, for a-C:H, tribo-patches form predominantly near the ends of the stroke while $T_f$ is larger at the middle of stroke (i.e. at higher sliding speeds) and regardless of the local temperature the hard ta-Cs undergo much more wear.

**Basic surface analysis**. After the friction tests, the discs and cylinders were cleaned twice in an ultrasonic bath with n-heptane (Chimie Plus: >99%) for 10 min to remove residual oil. The sample was handled with metallic tweezers without touching the surface of the disc. The wear volumes of the disc and cylinder were measured using an optical white light interferometer (Contour GT-K1, Bruker). The surfaces were observed without any conductive coating by SEM using a FEI XL30-FEG equipped with an Everhardt-Thornley secondary electron detector and operating under high vacuum. The acceleration voltage was set between 2 and 5 kV. Chemical composition analyses were carried out by EDX using an Oxford Instruments X-max silicon drift detector (80 mm² ultra-thin window). Quantitative analysis of the EDX spectra was performed using the Aztec software.

**X-ray photo electron spectroscopy**. XPS analysis was performed on the discs using an ULVAC-PHI Versa Probe II spectrometer equipped with a monochromatic Al Kα X-ray source with a beam diameter of 200 μm. The binding energy scale was calibrated with respect to the C1s photo-peak at a binding energy of 284.8 eV. The error of binding energies is estimated as ±0.1 eV. First, a survey spectrum was performed using a pass energy of 187.85 eV to identify all the elements present on the surface. Afterwards, narrower scans with a range of 20 eV were acquired using a pass energy of 23.5 eV to accurately identify the chemical state of each element and to perform quantitative analysis using PHI Multipack software. The contribution of the background was approximated by the Shirley method and Wagner sensitivity factors (corrected for the transmission function of the apparatus) were used for the calculation of the atomic concentrations. An Argon ion beam was used to sputter the DLC surfaces at an angle of 45°. The ion beam was accelerated at 250 V with a current density of 120 nA mm⁻². We sputtered the surfaces for 5 min. The corresponding sputter depth is ~0.5 nm[28].

**FIB-TEM analysis**. Dual Beam Focused Ion preparation was performed on a thin cross-section of the wear track of the disc employing Manutech-USD. A platinum gaseous precursor was first used to deposit a platinum layer onto the DLC surface, to protect it during milling and avoid any re-deposition. This layer was deposited in two steps. First, a low-energy electron beam was used to avoid damaging the DLC surface. Afterwards, a Ga ion beam was used at high energy to increase the growth rate of the protective platinum film and accelerate the process. The a-C:H thin foils obtained were analysed by TEM using a Jeol 2010F-UHR TEM equipped with a Schottky field emission gun operating at 200 kV. Bright-field images were acquired using a Gatan Orius 100 CCD camera and HAADF images in the Scanning Transmission Electron Microscopy (STEM) mode. EDX (80 mm² silicon drift detector from Oxford Instruments) was used for local chemical analysis (spectra or elemental maps). The ta-C(66) thin foil was analysed using a FEI Titan ETEM G2 electron microscope operated at 300 keV and equipped with a Cs image aberration corrector. To avoid contamination prior to the analysis, the samples were plasma-cleaned with Argon for 30 s. Compositional analyses were conducted in TEM mode with an energy dispersive X-ray spectrometer (SSD X-max 55 mm² from Oxford Instruments).

**Contact mechanics calculations**. Elastic-plastic contact of two rough DLC surfaces were modelled by using experimental AFM topography maps with discrete height profiles $h_{x,y}$ over a scan area of $20 \times 20$ μm² (including $512 \times 512$ data points). Local contact pressures $p_{x,y}$ and elastic displacements $u_{x,y}$ were calculated under an external normal pressure of 210 MPa (equal to a maximum cylindrical Hertzian contact pressure used in our experiments) using the PyCO code

developed by Pastewka and co-workers. Plasticity of surface asperities was taken into account if the local contact pressure $p_{x,y}$ went beyond the material hardness[50]. However, no significant differences in $P_{l,eff}$ were observed between elastic-plastic and full elastic calculations. The details of numerical techniques used in our contact mechanics calculations are described somewhere else[51–55]. Young's moduli and hardness values are tabulated in Table 1. A typical value of 0.2 was used for the Poisson's ratio of DLC. For each pair, 25 contact mechanics calculations were carried out by moving a topography with respect to the other along the $x$ axis with a step of 20 pixels. We assumed that the load is entirely supported by asperities and no fluids are entrained in the contact zone. Thus, $P_{l,eff}$ is only valid for boundary lubrication, i.e. at the ends of the stroke. We employed AFM topographies measured inside the wear tracks after running-in since they are representative of DLC surfaces in a steady, low-friction state. A contact mechanics calculation of DLC topographies measured outside wear tracks showed that significant amounts of surface asperities undergo plastic deformation (corresponding to an initial high friction in the Fig. 1b). However, the high friction state lasts only for the first few hundred cycles and $\mu$ drops immediately. For example, for a-C:H, the RMS heights and slopes outside the wear track are 2−3 times larger than those inside the wear track after running-in, which results in a much larger $P_{l,eff}$, and no correlation between $P_{l,eff}$ and $E^* h'_{rms}$ was found.

**Quantum chemical calculations.** Bond-breaking and -forming of ZDDP in contact with DLC surfaces were studied using third-order density-functional tight-binding (DFTB3) calculations[30] as implemented in the Atomistica software suite[56]. Atomic forces are extracted from quantum chemical calculations in order to describe complicated bond-breaking and -formation processes under tribological conditions. The accuracy of Slater–Koster parameters for modelling the interactions between ZDDP and DLC surfaces is confirmed by first-principles density-functional theory (DFT) calculations (Supplementary Fig. 10).

In this study, three types of DLC surfaces were considered: a-C:H with the density $\rho = 2.0\,\mathrm{g\,cm^{-3}}$ and hydrogen content $C_H = 20\,\mathrm{at\%}$, a-C with $\rho = 2.0\,\mathrm{g\,cm^{-3}}$, and ta-C with $\rho = 3.0\,\mathrm{g\,cm^{-3}}$. The aims of the simulations in Figs. 5 and 6–8 are different. In Fig. 5, the bulk shearing of sulphur-doped DLCs aims to understand weakening of the bulk regions under shear. Thus, the three computational models were chosen to exactly correspond to experimental a-C:H, a-C, ta-C(78), respectively. In contrast, in the simulations of Figs. 6–8, we considered the former two DLC models due to the following reasons. Our focus is on understanding tribochemical reactions of ZDDP on DLC surfaces. In general, hydrostatic compression of a-C increases its density and $sp^3$ percentage in the bulk and as a result forms ta-C with a high $sp^3$ percentage $p_{sp^3}$[38]. However, a superimposed shear motion drives the system away from its equilibrium phase. At the sliding interface, a ta-C surface layer undergoes $sp^3$-to-$sp^2$ rehybridisation resulting in a significant decrease of $p_{sp^3}$[33]. This tribolayer with $p_{sp^3} \approx 10\%$ should be universal for any hydrogen-free DLC coatings and therefore we modelled the reactive zone on top of the ta-Cs as a low-density a-C.

All DLC samples were generated by quenching a melt from 5000 to 0 K at a constant rate of $1\,\mathrm{K\,fs^{-1}}$. For Fig. 5, sulphur-doped ta-C samples with initial densities $\rho = 2.0$ and $3.0\,\mathrm{g\,cm^{-3}}$ and a same initial cell size of $14 \times 14 \times 14\,\mathrm{Å^3}$ were produced as starting configurations. For the DLCs in Figs. 6–8, three samples with a cell size of $15 \times 15 \times 10\,\mathrm{Å^3}$ were generated. For each sample, three surfaces were then created by cutting them perpendicular to the $z$-axis at different $z$ coordinates, separating the resulting slabs by 2 nm and introducing a ZDDP molecule (with ethyl groups) into the created vacuum region. Before inserting the ZDDP molecule, the DLC slabs were fully relaxed. Thus, in total, nine independent quasi-static molecular statics contact-closing/reopening trajectories were generated for each type of DLC. During QMS simulations, the cell size $L_z$ in the z-direction is decreased with a step $\triangle L_z = -0.2\,\mathrm{Å}$ and once the system reaches a desired contact pressure $P_0$, $L_z$ is increased to the initial one ($L_0 = 30.0\,\mathrm{Å}$) in steps $\triangle L_z = +0.2\,\mathrm{Å}$. For each step, the entire system is relaxed so that all force components are below $5.0 \times 10^{-3}\,\mathrm{eV\,Å^{-1}}$ with the FIRE algorithm[57]. This quasi-static contact-closing/opening simulation mimics an experimental situation in which two asperities come into contact and then detach from each other in the limit $v \rightarrow 0\,\mathrm{m\,s^{-1}}$. However, the effect of shear stress is not explicitly taken into account within this quasi-static simulation.

MD shearing simulations of two systems (ZDDP/DLC and sulphur-doped ta-C) were carried out with Lees-Edwards boundary conditions[31]. The system temperature $T$ was kept constant at 300 K using a Peters thermostat[58], and the equations of motion were integrated with a time step $\Delta t = 0.5\,\mathrm{fs}$ using the velocity-Verlet algorithm[59]. The system pressure was controlled using a Berendsen barostat[60]. The systems were sheared at a constant speed along the $x$ axis, and the averaged shear stress $\tau$ was calculated from the component $\tau_{zx}$ of the stress tensor.

Although boundary conditions in our atomistic simulations were matched to local experimental conditions (e.g., temperature and local contact pressures), the sliding speed of $v_{MD} = 30\,\mathrm{m\,s^{-1}}$ in MD simulations is two orders of magnitude larger than the experimental values ($v_{exp} = 0.157\,\mathrm{m\,s^{-1}}$), as in previous MD studies[61–63]. The use of such high sliding speeds is necessary to simulate a long sliding distance, sufficiently sample phase space, and especially describe tribochemical reactions and subsequent phase transitions at sliding interfaces[2,61].

However, as long as the sliding speed is well below the speed of sound in solids[64], the heat generated at sliding interface can be rapidly dissipated to surrounding bodies (which is often modelled by coupling the thermostat to the system). This is a rationale for reliable MD modelling of friction of materials. In our study, $v_{MD}$ is two to three orders of magnitude smaller than the speed of sound in DLCs. Indeed, we did not observe any artefacts of the higher sliding speed on the motion of atoms.

In addition, when the temperature is low enough, thermal activation of chemical reactions is negligible. This means that the chemical reactions of ZDDP are dominated by the competition between the mechanical strengths of the different chemical bonds and independent of the sliding speed. Thus, the observations in our MD simulations should be transferable to our low-speed experiments. This argument is confirmed by the observation that our MD shearing simulations reveal similar results as our quasi-static calculations.

## Data availability

Source data generated in this study are provided in the Source Data file. Data generated in this study have been deposited under the Zenodo repository at https://doi.org/10.5281/zenodo.4898983. Source data are provided with this paper.

## Code availability

The density-functional tight-binding calculations were performed using the open-source software Atomistica. The source code is available under GNU GPL v2 at http://www.atomistica.org. The density-functional theory calculations were carried out using the open-source software CP2K. The source code is available under GNU GPL v2 at https://www.cp2k.org. The contact mechanics calculations were carried out using the open-source software PyCo. The source code is available under MIT license at https://github.com/ComputationalMechanics/ContactMechanics.

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

## Acknowledgements

The authors gratefully acknowledge funding by the BMWi within the project PROMETHEUS. Part of the simulations was performed on the computational resource ForHLR II funded by the Ministry of Science, Research and the Arts Baden-Württemberg and DFG ("Deutsche Forschungsgemeinschaft"). Computing time was granted by the John von Neumann Institute for Computing (NIC) and provided on the supercomputer JUWELS[65] at Jülich Supercomputing Centre (JSC) within projects HFR04 and HFR09. The authors acknowledge the Consortium Lyon Saint-Etienne de Microscopy (CLYM) for the access to the electron microscopes and funding by the European Union within the Auvergne-Rhône-Alpes region FEDER project, PLATEFORME TRIBOLOGIE MOTEURS.

## Author contributions

J.M.M., M.M. and M.I.D.B.B. conceived and supervised the research. V.R.S carried out the tribological experiments. V.R.S. and J.G. conducted the XPS analysis. V.R.S., M.B.H. and K.M.M. performed the electron microscopy observations. C.H. designed the experimental conditions for the sliding tests and made mechanical modifications to the tribometer. T.K., G.M. and M.M. designed the computational work. T.K. performed the density-functional tight-binding calculations. S.M. performed the contact mechanics calculations. L.M. performed the density-functional theory calculations. V.R.S., T.K., J.M. M., G.M., M.M. and M.I.D.B.B. interpreted the findings and wrote the manuscript, with inputs from the other authors. All authors discussed the results and approved the final version of the manuscript.

## Funding

## Competing interests

The authors declare no competing interests.

**Additional information**

