## [Peer Review File · Nature Communications]

REVIEWER COMMENTS

Reviewer #1 (Remarks to the Author):

This paper reports extensive experimental analyses with contact-mechanics/quantum-chemistry modeling of ZDDP-lubricant effects on friction and wear of DLC coatings. The experimental data and the modeling results are valuable in understanding lubrication mechanisms of ZDDP in friction and wear of soft (a-C:H, a-C, and ta-C(51)) and hard (hard ta-Cs) DLC coatings. Nanoscale effective asperity-contact pressure is estimated with Persson's contact mechanics theory, and it is found that the dissociation of ZDDP under low effective contact pressure ($\lesssim 3$ GPa) of the soft DLCs leads to the growth of an anti-wear tribofilm consisting of 364 zinc polyphosphates. In contrast, high effective contact pressure (> 9 GPa) of the hard DLCs promotes ZDDP fragmentation, which weakens the carbon matrix through mechanochemical sulfur doping and leads to massive wear of ta-Cs. In addition to the contact pressure effect, the influence of contact shear traction and hydrogen on the tribofilm formation is revealed. Based on the new findings, publication of this article in Nature communications is recommended.

Minor comments to improve the presentation:

Could you provide an estimation of local temperature near the frictional sliding region?

Could you comment on the phase stability of the amorphous carbon at the high contact pressure?

What is the radius of the contacting cylinder of the tribological test?

Do you mean the Hertzian contact in your text a 2D cylindrical contact instead of a 3D spherical contact?

Reviewer #2 (Remarks to the Author):

Review of manuscript number NCOMMS-20-49225 entitled "Interplay of mechanics and chemistry governs wear of diamond-like carbon coatings interacting with ZDDP-additivated lubricants" that has been submitted for publication in Nature Communications.

This paper investigates the tribochemistry of a common antiwear additives in automotive lubricants, zinc dialkyl dithiophosphate, ZDDP, on various diamond-like carbons (DLC) with different mechanical properties and topographies. DLC's are used as coating for engines and, while ZDDP was originally used with conventional ferrous surfaces, this is an ubiquitous oil additive and will therefore be used in DLC-coated engines, and is thus of technical interest. The paper indicates that ZDDP was designed for ferrous surfaces – in fact, it was included as an antioxidant and its antiwear properties were found by completely by accident.

As a general comment, this paper exemplifies the problems associated with trying to identify and understand the chemical processes occurring at sliding interfaces where one is forced to try to infer what happened in the contact from an analysis of the resulting film. In this case, in addition to friction and wear measurements, the surface and subsurface region were analyzed by XPS, TEM and EDX

analyses of FIB-prepared samples. Sophisticated molecular dynamics (MD) simulations were carried out to investigate the chemical processes occurring at the interface and to relate them to the measured wear rates and surface analyses. However, it is not clear to me that the correspondence between the two is very strong and would lead me to be convinced that it faithfully reproduced what is occurring in the experiment. This is in part because the conditions used for the MD simulations, which require very high sliding speeds, and are very different from those in the experiment. In addition, the accuracy of the predictions depend on the accuracy of the potentials and, while this group is extremely competent at carrying out such simulations, small errors in energy barriers can translate into large differences in relative reaction rates.

One of the central conclusions is that the chemistry is controlled by the local asperity contact pressure $P_{l,eff}$ calculated from a theory due to Persson, and depends on the Young's modulus and root-mean-square (RMS) slope h'_{rms} of the DLC coating.

The RMS slope is supposed to be calculated from the root of the sum of the squares of the data on the disk and the cylinder (Line 348), but if I take the data presented in Table S2, there seem to be differences between the values calculated using this formula and those reported in Table 1. For example, the values calculated for the five samples from the data in Table S2 are 0.072, 0.15, 0.077, 0.098 and 0.103, while that values reported in Table 1 are 0.05, 0.102, 0.039, 0.067 and 0.101. Furthermore, when I estimate the range of errors, they vary from ~10% up to ~60%. This is evident from the differences between the values measured on the cylinder and disk. Given the centrality of $P_{l,eff}$ in their conclusions this should be checked and the effect of the wide range in this value noted and discussed.

In addition, more detail should be given on how the slopes were measured? What step sizes were used and how did it influence the resulting value? In addition, the standard deviation should be given on the individual values as well as the composite value.

The simulations also suggest that the mechanical properties of the film is influenced by penetration of sulfur into the bulk that influences the wear rate of the film and that the mechanical properties of the carbonaceous substrate are influenced by the diffusion of sulfur into the subsurface of the sample. This is an interesting idea although no explanation is given as to why this should occur or what might be the driving force. However, there seems to be no evidence for this occurring. Perhaps an EDX map of a FIB sample might have shown evidence for this process, but a sulfur map was not shown in Fig. 4 and, in any case, they analyze an a-C:H sample, where the simulations show a modest penetration of 4 Å, while is penetrated much further in the a-C sample. Carrying out similar measurements for an a-C sample and measuring the sulfur concentration below the wear pad would have provided strong evidence that the MD simulations were indeed accurately mimicking the experiment.

Fig. 2a: How was the composition measured for the XPS data? Scale bars for the counts should be included in Fig. 2a to indicate that the signals outside the track are smaller than those inside. The spin-orbit split components should be plotted as a single profile for clarity. How was the sample handled in

moving from the tribometer to the XPS machine.

Line 273: I am surprised that the reaction is initiated by Zn-S bond cleavage. While there is not much data on the way in which ZDDP decomposes, the little data available on steel surfaces suggests that it occurs via an initial interaction with the alkyl group. Was any phosphorus detected? Only the sulfur data are shown. Table 2 indicates that there is as much P and S in the films.

Line 284, Fig. 6g: The linear dependence of pressure is surprising here – an exponential increase would be expected.

Line 288: The XPS results in Fig. 2d indicate that Zn-S bond formation dominates in most cases, not S-C bond formation as suggested here.

Line 408: Define HEF IREIS. I assume that it is a company.

Figure S2: There seems to be a significant difference in the P binding energy. Is this fluctuation in the measurement or real shifts. If it is the latter, what is it due to?

In summary, these are interesting, but rather arcane results that are of interest to the tribochemical community, but it is not clear to me that they will be of general interest to the scientific community in general as it does not describe a new physical concept or chemical idea.

Reviewer #3 (Remarks to the Author):

In this manuscript, Ruiz et al. use experiments, contact mechanics calculations, and ab initio molecular simulations to study the friction and wear of diamond-like carbon (DLC) combined with lubricants containing zinc dialkyldithiophosphate (ZDDP).

The reciprocating tribometer experiments show that ZDDP generally has antagonistic effects on the friction and wear of DLC surfaces. This can be mitigated somewhat by tuning the hardness of the DLC. The hydrogen content and roughness of the DLCs also have a smaller effect on tribological performance with ZDDP-containing lubricants. Relatively low friction and wear can be achieved using a DLC with intermediate hardness. It should be noted, however, that the presence of ZDDP in the PAO base oil has a negligible effect on tribological performance in this case. Softer coatings exhibit similarly low wear and thin ZDDP-derived patchy tribofilms but higher friction. Conversely, harder ta-Cs undergo severe wear and sub-surface sulphur contamination. Contact-mechanics and quantum-chemical simulations suggest that shear combined with the high local contact pressure caused by the contact stiffness and average surface slope of hard ta-Cs favour ZDDP fragmentation and sulphur release. In absence of hydrogen, this is followed by local surface cold welding and sub-surface mechanical mixing of sulphur resulting in a decrease of yield stress and wear.

Overall, the methods are robust, the manuscript is well written, and results presented are certainly of interest to the tribology community. In places, the manuscript is difficult to follow, and the overall length of the manuscript probably needs to be reduced for this journal. I also have some doubts as to whether the work presented is of sufficient general interest for this journal, although I will leave this to the editor's discretion. My main technical issue is that none of the systems tested show a synergistic DLC-ZDDP combination.[1] At best, there is a no-harm relation between DLC and ZDDP – as noted for DLCs with moderate hardness. Thus, although ZDDP forms a thin tribofilm, it provides negligible additional wear protection over pure base oil under the studied conditions.

Specific questions, comments and recommendations are given below.

1. Page 2: These types of lubricant additives are typically called friction modifiers (FMs)[2] rather than anti-friction (AF).
2. Page 3: Recent QCM experiments have shown that ZDDP adsorption is much weaker on DLC surfaces compared to steel.[3] Since adsorption is a necessary first step for mechanochemical decomposition and ultimately tribofilm formation (Ref. 2 in the manuscript), this goes some way to explaining the thinner tribofilms formed on DLC compared to steel.
3. Page 4: The overall aim of the study is to provide surface-lubricant synergies, as reviewed by Neville et al.[1]
4. Page 5: It is worth noting that 1 wt.% ZDDP is similar to that used in commercial formulations (Ref. 15).
5. Page 5: Table 1 – It would also be useful to have a column showing estimates of the sp²/sp³ ratio, which can be correlated to the hardness.[4] It would also be useful to explain in this part of the text that DLCs with a higher sp²/sp³ ratio (more diamond-like) are generally harder.
6. Page 5: Table 1 – It would also be useful to have a column showing the density of the different DLCs[5] for comparison with the ab initio simulations.
7. Page 5: Table 1 – The ZDDP molecules may decompose even during the running-in period. It would thus also be good to have estimates for contact pressure using the initial RMS roughness and slope as well as after running in. Additional (dotted) lines could be added to Figure 1e to show the change in pressure distribution for each surface before/after running in.
8. Page 6: Figure 1a/b – rubbing time or sliding distance are probably more useful than cycles on x-axis of (a) and (b). Perhaps one of these could be added as a second x-axis at the top of the plots.
9. Page 6: Figure 1a/b – In rolling/sliding steel-steel contacts, ZDDP increases friction compared to pure base oil due to the formation of thick (ca. 100 nm) tribofilms that inhibit fluid entrainment into the contact, which means that the system enters the mixed/boundary regime at lower entrainment speeds.[6] Since this study uses a reciprocating tribometer, the system probably remains in the boundary regime for all of the studied systems.
10. Page 6: Figure 1a/b – none of the systems tested show a synergistic DLC-ZDDP combination.1 At best, there is a no-harm relation between DLC and ZDDP – for DLCs with moderate hardness. Thus, although ZDDP forms a thin tribofilm, it provides negligible additional wear protection over pure base oil under the studied conditions. It should be noted that the type of tribometer tests chosen (reciprocating cylinder-on-disk) were originally developed as a fuel lubricity test and only representative of a limited number of components, such as the piston ring-cylinder liner. DLC-ZDDP synergies could perhaps be

uncovered using a different type of tribology test, e.g. a rolling/sliding ball-on-disk tribometer (e.g. Ref. 5-7), which is representative of more components (e.g. bearings, gears, cam-follower systems).

11. Page 6: Figure 1a/b – It should be noted in the text that the presence of ZDDP in the PAO base oil has a negligible effect on tribological performance for the intermediate hardness DLC.

12. Page 6: Figure 1c – These results should be discussed in the context of the Archard wear law,[7] which predicts a linear increase in wear with material hardness. Clearly, this equation for abrasive wear does not hold for these systems (particularly in the presence of ZDDP). For PAO only systems, wear could follow a stress-promoted thermal activation model, as for nanoscale DLC contacts.[8] This could be tested by varying the load (and thus stress) in the tribometer experiments, perhaps for a subset of systems. For the case with ZDDP, wear is much more severe and is clearly driven by tribochemical processes.

13. Page 6: Figure 1e/f – these figures assume no fluid entrainment (i.e. boundary not mixed lubrication regime) which is only the case at the ends of the stroke (Figure S8), this should be noted either in the text or figure caption.

14. Page 7: Previous studies have shown similar trends, e.g. Ref. 5 showed that wear was greater for harder ta-C than softer a-C:H. The results of the current study suggest that friction and wear are similar for ta-C and a-C:H with similar hardness, suggesting that the differences observed in Ref. 5 were driven mainly by hardness rather than chemical effects due to the presence of H atoms.

15. Page 10: The observed tribofilm thickness is similar to that observed in previous experimental studies e.g. Ref. 7 in the manuscript.

16. Page 12: Figure 5 - Which of the experimental systems do these densities ($2/3 \text{ g cm}^{-3}$) correspond to? This information should be added, but may fit better in the Methods section on Page 24.

17. Page 13: The surfaces cannot be 'unterminated' – I think the authors mean they are not hydrogenated.

18. Page 15: In the simulations, the authors could have studied the effects of surfaces with different sp^2/sp^3 ratios to represent the different DLC hardness values.[4] This could significantly affect reactivity with ZDDP.

19. Page 16: I find it hard to justify the observation that hydrogenation plays no role in ZDDP decomposition kinetics. Many previous experiments have shown that additive reactivity is very different between these surfaces. The authors suggest that this entirely driven by differences in hardness and topography, rather than chemistry. However, surely the H atoms must passivate the surface C atoms to some degree?[9] Could the similar reactivity be due to the very large compression rates used in the current ab initio simulations?

20. Page 18 (and 23): All of the contact mechanics work assumes that the load is entirely supported by the asperities and the fluid plays a negligible role. Thus, all of the pressures are overestimated and only valid at the end of the stroke, where there is no fluid entrained. This should be acknowledged and justified in the text.

21. Page 19: The authors should be clear that the 'critical contact pressure' referred to here is for ZDDP decomposition.

22. Page 20: It should be noted here that previous studies (Ref. 14-15) have shown that ZDDP tribofilms can be formed with negligible asperity contact on steel and tungsten carbide surfaces by using lubricants that give very high shear stress. However, a more recent study showed that tribofilms cannot be formed

on DLC under similar conditions, probably due to the weak surface adsorption.[3]

23. Page 21: Since the conditions inside reciprocating tribometers are quite severe, the flash temperature rise should be estimated (e.g. Archard equation[10]). This will be similar for pure PAO and PAO+ZDDP but could vary significantly for the different ZDDPs, due to their differing thermal conductivities.

24. Page 21: Where were the ZDDP and PAO sourced from?

25. Page 21: It could be worth highlighting that the authors use a primary C4 ZDDP, which forms a tribofilm much more slowly than secondary ZDDPs (Ref. 15 in the manuscript) and is more thermally stable.[11]

26. Page 21: Which DLC hardness does this maximum Hertzian contact pressure correspond to?

27. Page 23: Were the surfaces relaxed before ZDDP molecules were added between them? Otherwise high surface energies are likely, making reactivity larger than would be the case for experimental surfaces.

28. Supporting Information: Typo – title of reference 3 should be: ‘A hybrid Gaussian and plane wave density functional scheme’

References

1. Neville, A., Morina, A., Haque, T. & Voong, Q. Compatibility between tribological surfaces and lubricant additives - How friction and wear reduction can be controlled by surface/lube synergies. *Tribol. Int.* 40, 1680–1695 (2007).
2. Spikes, H. Friction Modifier Additives. *Tribol. Lett.* 60, 5 (2015).
3. Ueda, M., Kadircic, A. & Spikes, H. ZDDP Tribofilm Formation on Non-Ferrous Surfaces. *Tribol. Online* 15, 318–331 (2020).
4. Savvides, N. & Bell, T. J. Microhardness and Young’s modulus of diamond and diamondlike carbon films. *J. Appl. Phys.* 72, 2791–2796 (1992).
5. LiBassi, A. et al. Density, sp³ content and internal layering of DLC films by X-ray reflectivity and electron energy loss spectroscopy. *Diam. Relat. Mater.* 9, 771–776 (2000).
6. Taylor, L. J. & Spikes, H. A. Friction-enhancing properties of ZDDP antiwear additive: Part I-Friction and morphology of ZDDP reaction films. *Tribol. Trans.* 46, 303–309 (2003).
7. Archard, J. F. Contact and Rubbing of Flat Surfaces. *J. Appl. Phys.* 24, 981 (1953).
8. Liu, J. et al. Tribochemical Wear of Diamond-Like Carbon-Coated Atomic Force Microscope Tips. *ACS Appl. Mater. Interfaces* 9, 35341–35348 (2017).
9. Schall, J. D., Gao, G. T. & Harrison, J. A. Effects of Adhesion and Transfer Film Formation on the Tribology of Self-Mated DLC Contacts. *J. Phys. Chem. C* 114, 5321–5330 (2010).
10. Archard, J. F. The temperature of rubbing surfaces. *Wear* 2, 438–455 (1959).
11. Jones, R. B. & Coy, R. C. The Chemistry of the Thermal Degradation of Zinc Dialkyldithiophosphate Additives. *ASLE Trans.* 24, 91–97 (1981).

Point-by-point response to the reviewers' comments

We thank all three reviewers for their efforts to evaluate our manuscript. We feel honoured by their positive assessment of our work and appreciate the time they invested for their in-depth inspection of our manuscript. The high number of useful comments is extraordinary and lead to a substantial improvement in the presentation and discussion of our results in the revised version of our manuscript.

Reviewer #1

This paper reports extensive experimental analyses with contact-mechanics/quantum-chemistry modeling of ZDDP-lubricant effects on friction and wear of DLC coatings. The experimental data and the modeling results are valuable in understanding lubrication mechanisms of ZDDP in friction and wear of soft (a-C:H, a-C, and ta-C(51)) and hard (hard ta-Cs) DLC coatings. Nanoscale effective asperity-contact pressure is estimated with Persson's contact mechanics theory, and it is found that the dissociation of ZDDP under low effective contact pressure ($\lesssim 3$ GPa) of the soft DLCs leads to the growth of an anti-wear tribofilm consisting of zinc polyphosphates. In contrast, high effective contact pressure (> 9 GPa) of the hard DLCs promotes ZDDP fragmentation, which weakens the carbon matrix through mechanochemical sulfur doping and leads to massive wear of ta-Cs. In addition to the contact pressure effect, the influence of contact shear traction and hydrogen on the tribofilm formation is revealed. Based on the new findings, publication of this article in Nature communications is recommended.

We really appreciate that the reviewer recommends publication of the manuscript in Nature Communications. As detailed below, we addressed the reviewer's comments and accordingly revised the manuscript.

Minor comments to improve the presentation:

1. Could you provide an estimation of local temperature near the frictional sliding region?

Interestingly, this question was not as easy to answer as we first thought. In our experiments we have a finite rectangular contact of two steel bodies coated with thin DLC layers. The standard flash temperature estimates in the tribology literature assume homogeneous bodies and report formulas for circular, quadratric and infinite band contacts. We had to go back to the pioneering work of Jaeger [Jaeger, J. Proc. R. Soc. N.S.W., 76, 203 (1942)] to start from the fundamental equations and consider the additional coatings using the Tian-Kennedy correction [Tian and Kennedy, J. Tribol. 115, 1-9 (1993)]. As detailed in the following table the sliding induced temperatures rises under boundary lubrication stay below 5° for all coatings.

Label	Young's modulus (GPa)	Density (g cm ⁻³)	sp ³ C p _{sp³} (%)	Thermal conductivity K (W m ⁻¹ K ⁻¹)	Heat capacity C _p (J g ⁻¹ K ⁻¹)	Thermal diffusivity D (10 ⁻⁶ m ² s ⁻¹)	Maximum flash temperature T _{fmax,BL} (°C)
a-C:H	259	2.07	-	1.01	0.74	0.66	2.3
a-C	287	2.34	31	1.12	0.74	0.65	4.1
ta-C(51)	493	2.77	62	1.92	0.69	1.01	0.5
ta-C(66)	572	2.91	73	2.22	0.67	1.15	2.2
ta-C(78)	625	3.01	79	2.43	0.66	1.23	3.1

We thus conclude that our experimental results are not affected by the flash temperature. We added the results of the flash temperature calculations and a brief discussion to the Methods section.

2. Could you comment on the phase stability of the amorphous carbon at the high contact pressure?

It is known that hydrostatic compression of a-C increases its density and sp^3/sp^2 ratio (e.g. Moras, Phys. Rev. Mater. 2, 083601, 2018). In our case the situation is more complicated because at the same time a superimposed shear deformation drives the system away from its equilibrium phase. If the DLC is a ta-C, both counter bodies remain ta-C (under 10 GPa), but at the sliding interface an a-C layer with roughly 10% sp^3 forms [Kunze et al., Tribol. Lett. 53, 119 (2014)]. Since the same layer can be observed during the sliding of diamond crystals, we believe that this tribomaterial is universal for hydrogen-free DLC coatings and therefore we modelled the reactive zone on top of the ta-Cs as a low-density a-C. Although we pointed this out in the original manuscript, the reviewer's question left us with the impression that this was not clear enough and therefore we added an improved justification of our simulations setup to the "Quantum chemical calculations" section in Methods.

3. What is the radius of the contacting cylinder of the tribological test?

We think that the reviewer missed this information. The diameter of the cylinder is mentioned in the "Materials" section of the original manuscript. Maybe the reviewer looked for the keyword radius. Now we state that the radius is 5 mm in the "Materials" section of Methods.

4. Do you mean the Hertzian contact in your text a 2D cylindrical contact instead of a 3D spherical contact?

Yes, it is common practice (see e.g. Johnson's Contact Mechanics, Cambridge University Press) to call also a cylinder-disk-pairing a Hertzian contact. We rephrased "Hertzian contact" as "cylindrical Hertzian contact" in the revised manuscript.

Reviewer #2

Review of manuscript number NCOMMS-20-49225 entitled "Interplay of mechanics and chemistry governs wear of diamond-like carbon coatings interacting with ZDDP-additivated lubricants" that has been submitted for publication in Nature Communications.

This paper investigates the tribochemistry of a common antiwear additives in automotive lubricants, zinc dialkyl dithiophosphate, ZDDP, on various diamond-like carbons (DLC) with different mechanical properties and topographies. DLC's are used as coating for engines and, while ZDDP was originally used with conventional ferrous surfaces, this is an ubiquitous oil additive and will therefore be used in DLC-coated engines, and is thus of technical interest. The paper indicates that ZDDP was designed for ferrous surfaces – in fact, it was included as an antioxidant and its antiwear properties were found completely by accident.

We thank the reviewer for reminding us that ZDDP was not originally designed as an antiwear additive for ferrous surfaces. We changed the corresponding sentence in the introduction of the revised manuscript.

As a general comment, this paper exemplifies the problems associated with trying to identify and understand the chemical processes occurring at sliding interfaces where one is forced to try to infer what happened in the contact from an analysis of the resulting film. In this case, in addition to friction and wear measurements, the surface and subsurface region were analyzed by XPS, TEM and EDX analyses of FIB-prepared samples. Sophisticated molecular dynamics (MD) simulations were carried out to investigate the chemical processes occurring at the interface and to relate them to the measured wear rates and surface analyses.

However, it is not clear to me that the correspondence between the two is very strong and would lead me to be convinced that it faithfully reproduced what is occurring in the experiment. This is in part because the conditions used for the MD simulations, which require very high sliding speeds, and are very different from those in the experiment. In addition, the accuracy of the predictions depend on the accuracy of the potentials and, while this group is extremely competent at carrying out such simulations, small errors in energy barriers can translate into large differences in relative reaction rates.

We thank the referee for acknowledging our competence in atomistic simulations. As always, we tried to match our atomistic simulations to the local experimental conditions. For instance, temperature and local contact pressures are realistic. The local contact pressures represent typical values in our contact mechanics calculations for experimental topographies and relevant flash temperatures are smaller than 5° in boundary lubrication regions.

On the other hand, it is true that sliding speeds are two orders of magnitude larger in the simulations ($v = 30 \text{ m s}^{-1}$) than in the experiments, as it is the case in many other MD studies. The use of such high sliding speeds is necessary to simulate a long sliding distance and to sufficiently sample phase space. In our previous studies (e.g. Kuwahara et al., Phys. Rev. Mater. 2, 073606, 2018, and Kuwahara et al., Nat. Commun. 10, 151, 2019), we discussed the effect of the sliding speed on tribochemical reactions of lubricants.

In principle, one must keep the sliding speed well below the speed of sound in materials. The speed of sound in DLC is two or three orders of magnitude larger than our sliding speed (Bullen, J. Appl. Phys. 88, 6317, 2000). This ensures that the heat generated in the contact during sliding is rapidly dissipated from the sliding interface to surrounding materials. The criterion is a prerequisite for reliable modelling of the shear response of materials and in this sense our simulations are sound.

In addition, at low enough temperatures (when thermal activation of reactions can be neglected) the reaction products are entirely determined by the mechano-chemical competition of the mechanical strength of the different chemical bonds. In this case, mechano-chemical reactions observed in our MD simulations should also be independent of the sliding speed and are thus transferable to our low-speed experiments. This argument is confirmed by the observation that the results of our MD shearing simulations are similar to those of our quasi-static calculations. Finally, concerning the transferability of our simulation results to technological applications, we note that the maximum sliding speed of a piston in automotive engines is of the same order of the sliding

speed in our MD simulations, making our simulations relevant to realistic automotive applications. Thus, we believe that sliding speed is not a critical issue.

Regarding the accuracy of interatomic potentials, our MD simulations were performed with a quantum chemical method, i.e. the atomic forces are extracted from density-functional tight-binding quantum chemical calculations and not from a classical interatomic potential. The use of the quantum chemical method is necessary to describe complicated bond-breaking and -formation processes under tribological conditions. The accuracy of the tight-binding Hamiltonian has been verified by additional first-principle DFT calculations of the same DLC/ZDDP/DLC tribological system (see Supplementary Note 10).

Concerning the accuracy of reaction barriers: even DFT can misrepresent reaction barriers by fractions of an eV. However, in our case a superimposed shear stress lowers barriers continuously and therefore many reactions happen independent of temperature and barrier height.

In conclusion, based on these arguments, we trust that our MD calculations are meaningful and help providing the correct interpretation of the experimental results. We added a brief discussion of the relevance and accuracy of our atomistic calculations to the "Quantum chemical calculations" section in Methods.

One of the central conclusions is that the chemistry is controlled by the local asperity contact pressure $P_{i,eff}$ calculated from a theory due to Persson, and depends on the Young's modulus and root-mean-square (RMS) slope h'_{rms} of the DLC coating.

The local asperity contact pressures $P_{i,eff}$ were obtained from a numerical boundary element solution of two experimental DLC topographies under compression (in Fig. 1e and 1f). Here, the Young's modulus and hardness enter. Persson's theory is used later on to interpret the numerical findings. We added a clarifying sentence to the caption of Fig. 1 and the "Contact mechanics calculations" section in Results.

The RMS slope is supposed to be calculated from the root of the sum of the squares of the data on the disk and the cylinder (Line 348), but if I take the data presented in Table S2, there seem to be differences between the values calculated using this formula and those reported in Table 1. For example, the values calculated for the five samples from the data in Table S2 are 0.072, 0.15, 0.077, 0.098 and 0.103, while that values reported in Table 1 are 0.05, 0.102, 0.039, 0.067 and 0.101.

We checked the numbers in our paper carefully and confirmed that they are correct. For each DLC, the RMS slope in Table 1 was the average of all AFM topographies measured after running-in inside the wear tracks on both the cylinder and disk. However, we completely agree with the reviewer that we should show RMS values for cylinder and disk separately as well as composite RMS slopes defined as $h'_{rms} = \sqrt{(h'_{rms,cylinder})^2 + (h'_{rms,disk})^2}$ in Table 1. We modified Table 1 accordingly and now show the averages of the cylinders and the disks with standard deviations.

Values given in Supplementary Table 2 refer to the topographies shown in Supplementary Fig. 1. Since they are not the average over all our measured surfaces they cannot be used to calculate numbers in Table 1.

Furthermore, when I estimate the range of errors, they vary from ~10% up to ~60%. This is evident from the differences between the values measured on the cylinder and disk. Given the centrality of $P_{i,eff}$ in their conclusions this should be checked and the effect of the wide range in this value noted and discussed.

As already explained above, the local asperity contact pressures $P_{i,eff}$ have nothing to do with the RMS slopes calculated here since $P_{i,eff}$ was obtained from numerical contact mechanics calculations using the AFM topographies as direct inputs, and were not estimated using Persson's theory. Thus, our conclusions are not affected by the values. The RMS slopes are used later on for the qualitative interpretation of the relation between local contact pressures and topography. We compare ta-C(51) with a-C, whose slope is almost two times larger, in order to explain the smaller $P_{i,eff}$ in the numerical contact mechanics calculations. Even if we take into account the errors in the RMS slope, the a-C slope is still 80% larger than the ta-C(51) slope and therefore our interpretation is valid.

We added clarifying sentences about the use of the numerical boundary element method to the caption of Fig. 1 and the “Contact mechanics calculations” section in Results.

In addition, more detail should be given on how the slopes were measured? What step sizes were used and how did it influence the resulting value?

We agree that it would be useful to show more details of AFM measurements and statistics of AFM data. As described in the Supplementary Information, the $20\mu\text{m} \times 20\mu\text{m}$ AFM topographies were measured with a resolution of 512×512 pixels. Consequently, the discretization size of our data is about 39 nm – slightly above the diameter of the AFM tip (20 nm). Note that discretization smaller than the tip size is prone to mapping errors. In contrast, increasing the grid spacing would lead to the loss of small-scale features of the surface topography and thus to smaller local contact pressures. Since we are interested in nanoscale contact pressures, the spacing should be as small as possible without being biased by AFM errors (according to a recent review by Jacobs et al. [Surf. Topogr.: Metrol. Prop. 5, 013001 (2017)]). Besides, we want to know the RMS slope h'_{rms} of the surface rather than a RMS roughness h_{rms} . The RMS slopes were calculated numerically by a central finite difference stencil (see Eq. S2). While h_{rms} is dominated by power-law scaling behaviour of a power spectral density at a largest wavelength, h'_{rms} depends entirely on the surface structure at a smallest scale. In order to check the accuracy of the calculated RMS slopes, we performed an additional $5\mu\text{m} \times 5\mu\text{m}$ AFM measurement with 256×256 pixels for the ta-C(51) disk (corresponding to a step size of 19.5 nm). Similar RMS slopes for both scans indicate that the step size of 39 nm is sufficiently small for RMS slope calculations. We added this explanation to Supplementary Note 1.

In addition, the standard deviation should be given on the individual values as well as the composite value.

This analysis has been performed for several topographies on both cylinder and disk for each DLC. In order to show the scatter, the standard deviations were added to Table 1.

The simulations also suggest that the mechanical properties of the film is influenced by penetration of sulfur into the bulk that influences the wear rate of the film and that the mechanical properties of the carbonaceous substrate are influenced by the diffusion of sulfur into the subsurface of the sample. This is an interesting idea although no explanation is given as to why this should occur or what might be the driving force.

In the MD, we clearly identified the mechanism for sulphur penetration. The cold-welded a-C surface layer (on top of the ta-C) is plastically deformed during sliding and the sulphur is mechanically mixed into the bulk. This mechanical driving leads to a strong non-equilibrium state with low shear resistance that accommodates the shear deformation. Since this metastable material is higher in energy than the original material, sulphur penetration cannot be described by a thermodynamic driving force. We note that this phenomenon can be commonly found in the literature. An example is the shear-induced amorphization of diamond and silicon (see for instance Moras et al., Phys. Rev. Mater. 2, 083601, 2018), whereby the shear-induced amorphous material is thermodynamically less stable than the original crystal but its formation is necessary to accommodate the shear deformation and minimise the shear resistance of the sliding interface. We clarify this aspect in Discussion of the revised manuscript.

However, there seems to be no evidence for this occurring. Perhaps an EDX map of a FIB sample might have shown evidence for this process, but a sulfur map was not shown in Fig. 4 and, and in any case, they analyze an a-C:H sample, where the simulations show a modest penetration of 4 Å, while is penetrated much further in the a-C sample. Carrying out similar measurements for an a-C sample and measuring the sulfur concentration below the wear pad would have provided strong evidence that the MD simulations were indeed accurately mimicking the experiment.

We thank the reviewer for the fruitful suggestion. In order to experimentally validate the sulphur penetration into the ta-C's subsurface, we performed the two types of additional measurements: XPS surface analysis after 5-min sputtering and TEM-EDX analysis of wear particles smeared on a ta-C surface. Both show strong evidence of sulphur penetration into the ta-C subsurface and thus supports our computational model in Fig. 5.

For ta-C(66), S2p XPS spectra remain similar even after 5-min sputtering (corresponding to a sputter depth of about 0.5 nm) and thus show the presence of significant amounts of S-C bonds. On the contrary, for a-C:H no

S-C bond signal was found after 5 min sputtering. These indicate that sulphur released from ZDDP is transported into the subsurface under shear for hard ta-C. The new XPS results are added to Fig. 2 as panel c.

Our new TEM-EDX analysis of a ta-C(66) surface after sliding is also informative. We observed not only the formation of large grooves but also filling of a groove with a smeared layer of wear particles (see Fig. 4d-f). Inside the smeared layer, we found about twice as much sulphur as zinc (Table 2). In contrast, sulphur was not detected inside the native ta-C bulk. A magnified TEM image shows that the sulphur-containing regions have a sp^2 -rich structure in agreement with our simulations. In addition, some graphitic ordering can be observed. Berman et al. also showed that sulphur can diffuse into diamond nanoparticles and induce the formation of an onion-like structure. We added the TEM-EDX analysis for ta-C(66) to Fig. 4.

Fig. 2a: How was the composition measured for the XPS data? Scale bars for the counts should be included in Fig. 2a to indicate that the signals outside the track are smaller than those inside. The spin-orbit split components should be plotted as a single profile for clarity. How was the sample handled in moving from the tribometer to the XPS machine.

The sample was rinsed and washed in pure n-heptane and introduced into the XPS machine immediately after the cleaning step. The sample was handled with metallic tweezers without touching the surface of the disk. This is explained in the “Basic surface analysis” section of Methods.

We can modify the figure to plot $2p_{1/2}$ and $2p_{3/2}$ spin-orbit components as single profiles (see the figure below), but it is not common practice and could confuse readers who are experts in surface analysis techniques such as XPS. We carefully checked previous similar studies by other groups and confirmed that it is common practice to show two spin-orbit peaks separately (e.g. Soltanahmadi et al., *Appl. Surf. Sci.* 414, 41, 2017). As explained in the Methods section, the peak fitting was made with both spin-orbit profiles for s and p orbitals which determine the position of the binding energies. We thus think that it is better to leave both profiles as in the original figure to avoid confusion.

Figure. S2p spectrum with spin-orbit components as single profiles.

Quantitative analysis were performed using the PHI Multipack software (as stated in the “X-ray photo electron spectroscopy” section of Methods). The absolute value of the signal of counts per second ($c s^{-1}$) of different samples cannot be compared. Therefore, it is preferable not to include it in the figure as this could be misleading.

In articles where the authors decide to include the scale, it is often labelled as “arbitrary units”. The quantitative analysis is made by comparing the relative quantities of the detected elements of the same data point. We made a figure with the intensity scale (counts/sec) below. By comparing the scale one could wrongfully assume that there is more sulphur on ta-C(78) (300 c s^{-1}) than on a-C (250 c s^{-1}). Similarly, comparisons between the inside and outside of the same DLC cannot be made as they are different data points.

Figure. S2p spectrum with the intensity scales (c s^{-1}).

Line 273: I am surprised that the reaction is initiated by Zn-S bond cleavage. While there is not much data on the way in which ZDDP decomposes, the little data available on steel surfaces suggests that it occurs via an initial interaction with the alkyl group.

As far as we know, there are no atomistic simulation studies that showed ZDDP's decomposition on surfaces under tribological conditions. Mosey et al. studied thermal decomposition of an isolated ZDDP molecule using first-principles molecular dynamics simulations (see Mosey et al., *J. Phys. Chem. A* 107, 5058, 2003). One or two Zn-S bonds in the ZDDP molecule break immediately at high temperatures ($> 700 \text{ K}$). However, Zn-S bonds repeatedly break and form, and the Zn atom stays bound to either 2 or 3 S atoms. Instead, an alkyl group is ejected from the ZDDP molecule. This decomposition process is different from that on DLC surfaces reported in our manuscript. Here, the broken Zn-S bonds cannot reform (as they do in Mosey's work) because they are replaced by the formation of Zn-C bonds with the surface. This interesting difference between reactions in the gas phase and on DLC surfaces is noted in the "Quantum chemical calculations" section of Results.

Moreover, in order to check the reviewer's comment that alkyl group should react first with steel surfaces, we performed first-principles DFT quasi-static contact-closing calculations of ZDDP confined between two Fe_2O_3 (0001) surfaces (used as a model for an oxidized steel surface). As the system is compressed, sulphur and oxygen atoms in ZDDP first interact with the surfaces. A further compression leads to Zn-S bond cleavage and S-Fe bond formation. The chemical reactions and contact pressures that are necessary for the reactions are also quite similar to those on DLC systems (as shown in Fig.6 of the manuscript). The difference of the ZDDP's decomposition pathway between steel and DLC surface is very interesting, but is out of the scope of this study. This interesting result of ZDDP/ Fe_2O_3 interactions will be scrutinized more systematically and included in a future publication.

Figure: Quasi-static DFT calculations of ZDDP confined between two Fe₂O₃ surfaces. The pressure along the z-axis is applied by gradually decreasing the z-axis dimension L_z in the simulation cell.

Was any phosphorus detected? Only the sulfur data are shown. Table 2 indicates that there is as much P and S in the films.

Indeed, in Fig. 2b we chose to show only the sulphur S2p spectra because there is an important difference in the chemical state of the sulphur species between the different DLCs, as shown by the shift in their binding energy. The other spectra (C1s, O1s, P2p, and Zn2p3) are shown in Supplementary Fig. 3 because there are no significant differences between their binding energies and chemical states in the different DLCs. Figure 2a shows the quantitative results (including both phosphorus and sulphur) obtained by XPS, and that there is as much phosphorus as sulphur as the reviewer correctly pointed out.

The new Table 2 includes quantitative analyses made by EDX on a FIB sample for the a-C:H and ta-C(66). Since the phosphorus signal overlaps with the platinum one, we decided to show only the sulphur and zinc content in the regions of interest.

Line 284, Fig. 6g: The linear dependence of pressure is surprising here – an exponential increase would be expected.

Let's assume that for a finite temperature reaction with a single barrier E_b the reaction rate k can be described by a pressure-activated Arrhenius law with an activation volume Ω and a prefactor k_0 :

$$k = k_0 e^{-\frac{E_b - \Omega P}{kT}}$$

This means that reaction rates increase exponentially with pressure. Assuming that the crucial reaction is the breaking of S-Zn bonds and $n(t)$ represents the fraction of sulphur atoms bonded to Zn then

$$n(t) = e^{-k_0 e^{-\frac{\Omega P - E_b}{kT}} t}$$

Here, t is a typical time scale characteristic for the duration of the simulation. Consequently, the fraction of sulphur atoms bonded to carbon is $m(t) = 1 - n(t) = 1 - e^{-\alpha e^{\frac{\Omega P}{kT}}}$ with $\alpha = k_0 e^{-\frac{E_b}{kT}} t$. The typical pressure dependence is depicted in the following diagram (for $\alpha = 0.001$):

Note the levelling off at large P reflecting that all S-Zn bonds have been broken and all S atoms are bonded to C (i.e. $m = 1$). On the other hand, for small enough P an exponential dependence $m(t) = \alpha e^{\frac{\Omega P}{kT}}$ can indeed be expected.

However, the data in Fig. 6g are obtained for a quasi-static (i.e. $T = 0$ K) simulation. Here reactions are not thermally activated and reaction rates are not meaningful (since there is no time in quasi static calculations). In this case, the pressure dependence of the S-C bond formation is a consequence of the pressures required to break the S-Zn bonds by overcoming some critical force as well as to bring the sulphur atoms in close contact with reactive carbon atoms on the DLC surfaces (i.e. by flattening the molecule or its fragments). Since we consider amorphous surfaces and also an initial random orientation of the ZDDP molecule there is a distribution of pressures for the S-C bond formation which cannot be predicted analytically. The only way to describe this is by performing the atomistic pressurization simulations – as it has been done in the manuscript. Note however, that a levelling off should also be seen in our quasi-static calculations and therefore an exponential dependence over the whole pressure range is very unlikely.

Line 288: The XPS results in Fig. 2b indicate that Zn-S bond formation dominates in most cases, not S-C bond formation as suggested here.

There is a misleading sentence starting from “Interestingly, the contribution from ...” on lines 198–200. Since Zn-S bonds are present in ZDDP there is no Zn-S bond formation. Accordingly for the soft DLCs there is still ZDDP in the wear scar and Zn-S XPS signal is strong. The situation reverses for the hard ta-C where the intensity of the Zn-S signal is diminished and S-C bonds become dominant. This suggests that ZDDP stays intact for soft DLCs, whereas it decomposes into fragments for hard ta-Cs. We corrected the sentence and added a sentence about the ZDDP’s bonding state in the “Surface chemical analyses” section in Results.

Line 408: Define HEF IREIS. I assume that it is a company.

The reviewer is right. DLC samples used were produced by the company HEF IREIS. We added “the company” before “HEF IREIS”.

Figure S2: There seems to be a significant difference in the P binding energy. Is this fluctuation in the measurement or real shifts. If it is the latter, what is it due to?

The shift is very small, only about 0.1-0.2 eV so it’s very close to fluctuation in the measurement. A shift in the P binding energy could appear depending on the polymerization degree of the tribofilm. Higher polymerization degrees shift the P binding energy to higher values. However, this very small shift is not employed in the literature to investigate the polymerization state. Instead the O(1s) peak is analyzed, since it shows a stronger shift between the two types of oxygen with P-O bond.

Reviewer #3

In this manuscript, Ruiz et al. use experiments, contact mechanics calculations, and ab initio molecular simulations to study the friction and wear of diamond-like carbon (DLC) combined with lubricants containing zinc dialkyldithiophosphate (ZDDP). The reciprocating tribometer experiments show that ZDDP generally has antagonistic effects on the friction and wear of DLC surfaces. This can be mitigated somewhat by tuning the hardness of the DLC. The hydrogen content and roughness of the DLCs also have a smaller effect on tribological performance with ZDDP-containing lubricants. Relatively low friction and wear can be achieved using a DLC with intermediate hardness. It should be noted, however, that the presence of ZDDP in the PAO base oil has a negligible effect on tribological performance in this case. Softer coatings exhibit similarly low wear and thin ZDDP-derived patchy tribofilms but higher friction. Conversely, harder ta-Cs undergo severe wear and sub-surface sulphur contamination. Contact-mechanics and quantum-chemical simulations suggest that shear combined with the high local contact pressure caused by the contact stiffness and average surface slope of hard ta-Cs favour ZDDP fragmentation and sulphur release. In absence of hydrogen, this is followed by local surface cold welding and sub-surface mechanical mixing of sulphur resulting in a decrease of yield stress and wear.

Overall, the methods are robust, the manuscript is well written, and results presented are certainly of interest to the tribology community.

We thank the reviewer for carefully reading the manuscript and acknowledging the novelty and importance of this work. We are also grateful to the reviewer for providing a lot of valuable and constructive comments.

In places, the manuscript is difficult to follow, and the overall length of the manuscript probably needs to be reduced for this journal. I also have some doubts as to whether the work presented is of sufficient general interest for this journal, although I will leave this to the editor's discretion. My main technical issue is that none of the systems tested show a synergistic DLC-ZDDP combination.[1] At best, there is a no-harm relation between DLC and ZDDP – as noted for DLCs with moderate hardness. Thus, although ZDDP forms a thin tribofilm, it provides negligible additional wear protection over pure base oil under the studied conditions.

We are glad that the reviewer understood the main point of this manuscript. We describe in the introduction that ZDDP can hardly be avoided in machinery with ferrous surfaces. At the same time, its presence could affect the properties of DLC coatings and prevent their use. With this in mind, it is very good news that we were able to find a recipe to produce DLC coatings that can be used for such technical systems. Of course, a synergistic effect would have been even better, but it is well known that a different kind of lubricant chemistry is needed to lower friction of DLC surfaces (e.g. some organic friction modifiers in pure base oil – see Ref. 1-3 in the manuscript). We added this to the conclusion paragraph of the revised manuscript.

Specific questions, comments and recommendations are given below.

As detailed below, we addressed the reviewer's minor comments and accordingly made revisions to the manuscript.

1. Page 2: These types of lubricant additives are typically called friction modifiers (FMs)[2] rather than anti-friction (AF).

Thank you for pointing out the use of the inappropriate terminology. Indeed, MoDTC is known as a friction modifier additive. We corrected the corresponding sentences in the revised manuscript.

2. Page 3: Recent QCM experiments have shown that ZDDP adsorption is much weaker on DLC surfaces compared to steel.[3] Since adsorption is a necessary first step for mechanochemical decomposition and ultimately tribofilm formation (Ref. 2 in the manuscript), this goes some way to explaining the thinner tribofilms formed on DLC compared to steel.

Thank you for bringing this important manuscript to our attention. Indeed, this is an interesting observation and could be a mechanism that explains the formation of thinner tribofilms on DLC. As shown in the response to

Reviewer #2, ZDDP can physisorb on a Fe₂O₃ surface by the formation of O_{ZDDP}-Fe and S-Fe bonds (without bond breaking in the molecule). This physisorption state (~0.5 eV) was not observed on the DLC surfaces, where the reactions of ZDDP start with breaking of Zn-S and formation of S-C bonds. This may give us a hint for understanding weaker adhesion of a ZDDP-derived tribofilm on DLC. We referred to the work by Ueda et al. in the Introduction of the revised manuscript.

3. Page 4: The overall aim of the study is to provide surface-lubricant synergies, as reviewed by Neville et al.1

Although synergies between surface and lubricants are extremely important and usually observed on an empirical level, our goal is different. We want to show that there are optimum coating properties (i.e. intermediate hardness and small RMS slope) that show no antagonistic effects on DLC's friction and wear in PAO+ZDDP. Moreover, this study aims to understand the interplay between underlying multi-scale/physics mechanisms i.e. between nano-mechanics and chemistry. In order to avoid confusion and make the aim of this study clearer, we replaced the term "synergistic role" with "crucial role" in the Introduction of the revised manuscript.

4. Page 5: It is worth noting that 1 wt.% ZDDP is similar to that used in commercial formulations (Ref. 15).

Thank you for the suggestion. We added it to the "Materials" section in Methods.

5. Page 5: Table 1 – It would also be useful to have a column showing estimates of the sp²/sp³ ratio, which can be correlated to the hardness.[4] It would also be useful to explain in this part of the text that DLCs with a higher sp²/sp³ ratio (more diamond-like) are generally harder.

Of course, it is obvious for DLC experts that hardness anti-correlates with sp²/sp³ ratio (i.e. the higher the sp²/sp³ ratio the less diamond-like). However, we agree with the reviewer that adding this information to the manuscript would be useful for the general reader. According to Robertson and co-workers [Ferrari, Surf. Coat. Technol. 180, 190, 2004], for hydrogen-free DLC, the percentage p_{sp^3} of sp³ C and density ρ (g cm⁻³) are calculated from the following relationships with the Young's modulus E (GPa):

$$\begin{aligned}\rho(\text{g cm}^{-3}) &= 1.92 + 0.0137p_{sp^3}, \\ E(\text{GPa}) &= 478.5(p_{sp^3}/100 + 0.4)^{1.5}, \\ \rho(\text{g cm}^{-3}) &= 1.37 + [E(\text{GPa})]^{2/3}/44.65.\end{aligned}$$

For a-C:H [Casiraghi et al., Diam. Relat. Mater. 14, 1098, 2005], the relation between the density and Young's modulus is given by

$$\rho(\text{g cm}^{-3}) = 0.257 + 0.011(E(\text{GPa}) + 511)/44.65.$$

The Table below shows the calculated densities and sp³ percentages for all DLCs. We added sentences regarding the correlation between the sp³ percentage and density of our DLC coatings to the Methods of the revised manuscript, and also added these values to Table 1.

Label	Young's modulus E (GPa)	Density ρ (g cm ⁻³)	sp ³ C p_{sp^3} (%)
a-C:H	259	2.07	-
a-C	287	2.34	31
ta-C(51)	493	2.77	62
ta-C(66)	572	2.91	73
ta-C(78)	625	3.01	79

6. Page 5: Table 1 – It would also be useful to have a column showing the density of the different DLCs[5] for comparison with the ab initio simulations.

We agree that it is useful to add DLC's bulk density to the revised manuscript. As shown above, we estimated DLC's densities based on empirical formulae given in Ferrari, Surf. Coat. Technol. 180, 190 (2004) and Casiraghi et al., Diam. Relat. Mater. 14, 1098 (2005). We added the numbers to Table 1.

It has to be noted that on top of as-grown ta-C a nm-thick low-density a-C layer is found (as we already discussed it in the manuscript) and gets even thicker during sliding (see the answer to the 2nd comment of reviewer #1). The estimated high density of 3.0 gcm^{-3} for ta-C(78) is representative for bulk, not for surface. Thus, a simulation model with 3.0 gcm^{-3} would not represent an experimental DLC surface. We also explained this point more clearly in the "Quantum chemical calculations" section of Methods.

7. Page 5: Table 1 – The ZDDP molecules may decompose even during the running-in period. It would thus also be good to have estimates for contact pressure using the initial RMS roughness and slope as well as after running in. Additional (dotted) lines could be added to Figure 1e to show the change in pressure distribution for each surface before/after running in.

According to the reviewer's comment, we carried out AFM measurements outside the wear tracks on both the cylinder and disk for all five DLC coatings, and then performed contact mechanics calculations of experimental DLC topographies. The figure below shows a comparison of local contact pressure distributions obtained by numerical contact mechanical calculations between the "running-in" (left, Fig. 1e) and "as-grown" (right) surfaces for all five DLC coatings. Although the distributions change slightly after running-in, no significant differences are observed for a-C, ta-C(51), and ta-C(66). The effective local contact pressures $P_{l,eff}$ are also similar between running-in and as-grown topographies. In contrast, the local contact pressure distribution for as-grown a-C:H is different from that for a-C:H after running-in. There is a large peak at $P_z = 27 \text{ GPa}$ (corresponding to its hardness), meaning that many asperities deform plastically. This plastic events correspond to an initial high friction state (see the friction curve in Fig. 1b). However, the friction coefficient drops immediately and reaches steady-state friction after a few hundred cycles. The initial asperity collisions lead to surface smoothing and thus to a significant decrease in the local contact pressure. Although this is an interesting observation, the contact pressures after running-in are more representative of the steady-state low friction coefficient. Thus, we kept Fig. 1e as it is, and instead added a brief discussion about local contact pressures at the very beginning of the friction test to the "Contact mechanics calculations" section in Methods.

8. Page 6: Figure 1a/b – rubbing time or sliding distance are probably more useful than cycles on x-axis of (a) and (b). Perhaps one of these could be added as a second x-axis at the top of the plots.

We agree with the reviewer, and added a second x-axis with sliding distance to Fig. 1a and 1b.

9. Page 6: Figure 1a/b – In rolling/sliding steel-steel contacts, ZDDP increases friction compared to pure base oil due to the formation of thick (ca. 100 nm) tribofilms that inhibit fluid entrainment into the contact, which means that the system enters the mixed/boundary regime at lower entrainment speeds.[6] Since this study uses a reciprocating tribometer, the system probably remains in the boundary regime for all of the studied systems.

Thank you for the fruitful comment. According to our EHL film thickness calculations (see Supplementary Note 8), our systems are in boundary lubrication ($\lambda < 1$) during portions of the stroke, i.e. near the edges – see the result of lambda calculations in the answer to the 1st comment of Reviewer #1. The percentage of the boundary lubrication regime depends on DLC's surface roughness. For our a-C:H system, boundary lubrication applies for about 18% of the stroke. This is confirmed by the result that much more ZDDP-derived tribopatches form near the edges of the stroke. In contrast, the a-C and ta-C(78) system are in boundary lubrication during the entire stroke since both surfaces are rougher than the others. For example, for ta-C(78) wear events increase the surface roughness resulting in a decrease of the lambda value at the very beginning of sliding. Thus, the boundary lubrication zone increases gradually and spreads to the entire stroke. We added a brief discussion about the lubrication regime in our reciprocating tests to Supplementary Note 8.

10. Page 6: Figure 1a/b – none of the systems tested show a synergistic DLC-ZDDP combination. At best, there is a no-harm relation between DLC and ZDDP – for DLCs with moderate hardness. Thus, although ZDDP forms a thin tribofilm, it provides negligible additional wear protection over pure base oil under the studied conditions.

Yes, there is no synergistic effect, but we demonstrate here that very hard DLCs should be avoided and the use of ta-Cs with moderate hardness represents a good solution to avoid antagonistic effect of ZDDP. Our main focus in this study is to elucidate the interplay of mechanics and chemistry and give mechanistic insights that connect atomic-scale mechanochemistry with macroscopic phenomena.

11. It should be noted that the type of tribometer tests chosen (reciprocating cylinder-on-disk) were originally developed as a fuel lubricity test and only representative of a limited number of components, such as the piston ring-cylinder liner. DLC-ZDDP synergies could perhaps be uncovered using a different type of tribology test, e.g. a rolling/sliding ball-on-disk tribometer (e.g. Ref. 5-7), which is representative of more components (e.g. bearings, gears, cam-follower systems).

We thank the reviewer for this important remark. The focus of this manuscript is on combustion engines (e.g. automotive applications). Hence, the use of the cylinder-on-disk reciprocating test mimics for instance a piston-liner system. However, in the distribution part of an engine, on the cam-latch or cam-follower contact, there is a reciprocating movement. This is not pure sliding, but the drive speed drops to 0 at the edge of the follower or at the end of the stroke of the big skate. This is one of the reasons for wear. Indeed when the driving speed drops to 0, the surfaces come into contact. On a cam, there can be wear on both sides of the nose. This also corresponds to the area where the driving speed drops to 0 and therefore the cylinder-on-disk reciprocating test is also useful for this case. Of course, we agree that almost no tribometer is representative of real contacts in engines. To be a little more representative of the cam-follower contact, ring/plane can be explored with the sliding speed ranging from 0.6 to 2.6 ms^{-1} and contact pressures of about 500 MPa. The real power dissipated by friction is generally around several tens of W in applications, while it's a few tens of mW in our tribometer. Therefore, we cannot exclude that other testing conditions could result in synergistic effects.

12. Page 6: Figure 1a/b – It should be noted in the text that the presence of ZDDP in the PAO base oil has a negligible effect on tribological performance for the intermediate hardness DLC.

We already mention this on page 7 in the original manuscript: "Thus, ZDDP has no major impact on the friction and wear of softer DLCs but a catastrophic effect on those of harder ta-Cs." Maybe this was not clear enough. We changed the sentence to "Thus, ZDDP has a negligible effect on the tribological performance of softer/moderate-hard DLCs but a catastrophic effect on those of harder ta-Cs."

13. Page 6: Figure 1c – These results should be discussed in the context of the Archard wear law,[7] which predicts a linear increase in wear with material hardness. Clearly, this equation for abrasive wear does not hold for these systems (particularly in the presence of ZDDP). For PAO only systems, wear could follow a stress-promoted thermal activation model, as for nanoscale DLC contacts.[8] This could be tested by varying the load (and thus stress) in the tribometer experiments, perhaps for a subset of systems. For the case with ZDDP, wear is much more severe and is clearly driven by tribochemical processes.

We agree that only in the PAO systems wear can follow a stress-promoted model. According to the reviewer's suggestion, we performed PAO lubricated tests by increasing the normal force F_N from 23 to 50 N (corresponding to $P_{\text{Hertz}} = 210 \rightarrow 310$ MPa) and observed that mild wear starts to occur for hard ta-C(66).

Thus, it is expected that the wear volume increases with the normal force. We also performed another test by reducing the sliding speed to half of the original value and observed an increase in wear. However, these tests are part of a different study. As stated by the reviewer, the article's length is already quite extensive and our main focus of this study is on the tribochemical wear of DLCs in the presence of ZDDP.

14. Page 6: Figure 1e/f – these figures assume no fluid entrainment (i.e. boundary not mixed lubrication regime) which is only the case at the ends of the stroke (Figure S8), this should be noted either in the text or figure caption.

The reviewer is right. We explained it briefly in the "Tribological tests" paragraph of the Methods section, and added a detailed discussion to Supplementary Note 8. See also the answer to point 9.

15. Page 7: Previous studies have shown similar trends, e.g. Ref. 5 showed that wear was greater for harder ta-C than softer a-C:H. The results of the current study suggest that friction and wear are similar for ta-C and a-C:H with similar hardness, suggesting that the differences observed in Ref. 5 were driven mainly by hardness rather than chemical effects due to the presence of H atoms.

We believe the reviewer misunderstood one step in our mechanistic model. Hard coatings favour high local contact pressure leading to cold welding and chemical mixing at the sliding interface. However, hydrogen surface passivation prevents this from happening. For instance, a hard a-C:H (a so-called ta-C:H) would not show mixing of S into DLC, while a ta-C with similar hardness does (hence the high wear). Thus, the detrimental effect of ZDDP depends also on chemistry. We added Ref. 5 (Vengudusamy Tribol. Int. 44, 165, 2011) and this explanation to the Discussion section.

16. Page 10: The observed tribofilm thickness is similar to that observed in previous experimental studies e.g. Ref. 7 in the manuscript.

We agree that we should refer to a previous result by Vengudusamy et al. (Tribol. Lett. 51, 469, 2013), and thus added this reference to the "Scanning and transmission electron microscopy characterizations" section of the revised manuscript.

17. Page 12: Figure 5 - Which of the experimental systems do these densities (2/3 g cm⁻³) correspond to? This information should be added, but may fit better in the Methods section on Page 24.

As explained in the answer to the 1st comment of Reviewer #1, experimental a-C:H and a-C have densities close to 2.0 gcm⁻³, whereas experimental ta-Cs have bulk densities of about 3.0 gcm⁻³. In Fig. 5, we investigated weakening of the bulk DLC under shear and in presence of sulphur. Therefore, there is a clear correlation between experimental and computational values of DLCs densities. We added this explanation to the "Quantum chemical calculations" section of Methods.

18. Page 13: The surfaces cannot be 'unterminated' – I think the authors mean they are not hydrogenated.

We agree that 'unterminated' is not correct. Here, we meant that no hydrogen atoms were added after two surfaces were created in order to passivate newly formed surface dangling bonds. We rephrased 'unterminated' as 'unpassivated' in the revised manuscript.

19. Page 15: In the simulations, the authors could have studied the effects of surfaces with different sp²/sp³ ratios to represent the different DLC hardness values.[4] This could significantly affect reactivity with ZDDP.

Hard ta-C usually contains more sp³ carbon atoms in the bulk compared with soft a-C. However, the sp³ content in the top surface region is completely different from the sp³ content in the bulk of the material, and reactions happen in a sp²-rich a-C top layer. According to a previous molecular dynamics study by Kunze et al. (Tribol. Lett. 53, 119, 2014), sliding ta-C and even diamond produces a surface a-C region with a low sp³ content of about 10%. This result is supported by experiments in the same paper. Based on these results, our 9 a-C systems have low sp³ contents ranging from 2 to 9%. Larger sp³ contents near surfaces would not be representative of a tribological DLC surface. We explained this more clearly in the "Quantum chemical calculations" section in Methods.

Nevertheless, we agree with the reviewer that checking the reactivity of a sp^3 -rich DLC surface would be useful. Therefore, we performed quasi-static contact-closing/opening simulations for 9 ta-C systems with higher sp^3 contents (ranging from 48 to 67%) and a density of 3.0 g cm^{-3} . The figures below show that the average number of S-C bonds and sulphur released onto the surface and their pressure dependence are similar to those for a-C:H and a-C. This indicates that an increase in the sp^3 content on the DLC surface has no significant impact on chemical reactivity of ZDDP (especially S-C bonds formation and sulphur release). We added a brief discussion about the effect of DLC's sp^3 content to Supplementary Note 6.

Figure: Quasi-static contact-closing/opening simulations for a-C:H, a-C, and ta-C in contact with a ZDDP molecule.

20. Page 16: I find it hard to justify the observation that hydrogenation plays no role in ZDDP decomposition kinetics. Many previous experiments have shown that additive reactivity is very different between these surfaces. The authors suggest that this entirely driven by differences in hardness and topography, rather than chemistry. However, surely the H atoms must passivate the surface C atoms to some degree?[9] Could the similar reactivity be due to the very large compression rates used in the current ab initio simulations?

Indeed, we observe that ZDDP's decomposition kinetics is in general faster on a-C than on a-C:H, reflecting the passivating action of H that lowers the amount of available reaction sites. For example, as shown in Supplementary Fig. 6, in an extreme case where surface dangling bonds are fully passivated with hydrogen atoms, reactions are significantly inhibited even at higher contact pressures ($> 10 \text{ GPa}$). We see stronger tendency, for example, for S-P bond breaking and P-C bond formation under shear on a-C than on a-C:H (see plots below). However, variations of reactivity due to pressure are much larger than reactivity changes due to variation of the chemical composition. Moreover, one should bear in mind that experimental differences in reactivity might also be affected by pressure/hardness differences. Thus, we should say that ZDDP decomposition kinetics is "dominantly" driven by difference in hardness and topography, rather than chemistry. We changed the relevant sentences to reflect this observation.

Regarding the reviewer's second question about the effect of compression rates: in our quasi-static simulations (in Fig. 6) a compression rate cannot be defined since there is no time scale involved in the compression process. The upper DLC surface is moved in step downwards followed by the full relaxation of the entire system after each step.

Figure: Molecular dynamics shearing simulations of a-C:H and a-C in contact with a ZDDP molecule. Averaged numbers of breaking (a) S-P and (b) forming P-C bonds after opening the contact as a function of the contact pressure P_z . The error bars represent standard errors of the mean.

21. Page 18 (and 23): All of the contact mechanics work assumes that the load is entirely supported by the asperities and the fluid plays a negligible role. Thus, all of the pressures are overestimated and only valid at the end of the stroke, where there is no fluid entrained. This should be acknowledged and justified in the text.

Of course the conditions are more severe at the ends of the stroke and this can be clearly observed in our surface topographies (see 2D images of ta-C(78) topographies below) showing that larger scratches form at the edges of stroke. The percentage of the boundary lubrication regime over the stroke depends on DLC's surface roughness. According to our lambda analysis (Supplementary Fig. 9), we are in boundary lubrication for 18% of the stroke (for a-C:H), which is where the proposed reactions occur and also our simulations are valid. We agree that on the rest of the stroke we should have mixed lubrication where the fluid can play a role in supporting some of the load. In this sense, our contact mechanics calculations are an upper estimate of the local contact pressures, which are valid estimates for portions of the stroke (18% for a-C:H). For the other systems, boundary lubrication applies for more than 32% of the stroke. We clarify this point in the "Contact mechanics calculations" section of Methods.

Figure: Surface topographies of coating ta-C(78) (taken after 800 cycles) showing that larger scratches form at the edges of stroke. Black areas represent pores of virgin coating.

22. Page 19: The authors should be clear that the 'critical contact pressure' referred to here is for ZDDP decomposition.

Actually we do not refer to a critical pressure for ZDDP decomposition. The term "critical" we used here is misleading. We should say that $P_{l,eff}$ is consistent/comparable with local pressure values measured at DLC tip/steel single-asperity contacts where ZDDP-film growth was observed. We changed this in the Discussion section of the revised manuscript.

23. Page 20: It should be noted here that previous studies (Ref. 14-15) have shown that ZDDP tribofilms can be formed with negligible asperity contact on steel and tungsten carbide surfaces by using lubricants that give very high shear stress. However, a more recent study showed that tribofilms cannot be formed on DLC under similar conditions, probably due to the weak surface adsorption.[3]

Thanks you for pointing out this interesting aspect. Indeed, as the referee indicates, probably due to the weak surface adsorption on DLC, boundary lubrication conditions are needed to either form a ZDDP-tribofilm or to cause sulphur doping of the DLC tribosurface. In our reciprocating friction tests, the tribofilm forms predominantly at the edges of the stroke (where the contacting surfaces experience boundary lubrication). SEM images clearly show that tribofilm formation is strongly reduced at the centre of the stroke, i.e. under mixed lubrication (see the figure below). We cited the interesting paper by Ueda et al. [3] in the Introduction, and added the SEM images and a brief discussion to Supplementary Note 4.

Figure: SEM images in SE detection mode of ta-C(51) (top) and a-C:H (bottom) at the edge (left) and center (right) of the wear track.

24. Page 21: Since the conditions inside reciprocating tribometers are quite severe, the flash temperature rise should be estimated (e.g. Archard equation[10]). This will be similar for pure PAO and PAO+ZDDP but could vary significantly for the different ZDDPs, due to their differing thermal conductivities.

Thank you for the important comment. We think the reviewer meant “different DLCs” rather than “different ZDDPs”. The Archard equation is applicable for circular contacts, e.g. a ball-on-disk configuration. For line contact of cylindrical surfaces, Jaeger’s formula has to be used. Moreover, since a DLC thin film is deposited on a steel substrate, properties of both materials have to be taken into account. In the answer to the first comment of Reviewer #1, we have shown results of flash temperature calculations as well as EHL film thickness (λ) calculations. For a-C (the coating with the lowest thermal diffusivity), the average flash temperature rise under boundary lubrication $T_{\text{fmax,BL}}$ is 4.1 °C. $T_{\text{fmax,BL}}$ for a-C:H is 2.3 °C since the BL regime accounts for only 18% of the stroke, i.e. near the edges at low sliding speeds. For ta-C(66) and ta-C(78) (the coatings with the highest thermal diffusivities), we obtain similar flash temperature rises of 2.2 °C and 3.1 °C, respectively. Therefore, $T_{\text{fmax,BL}}$ is negligible for all DLC coatings, and we find no clear correlations of the flash temperature with DLC’s properties and observed tribological phenomena. Indeed, we measure the highest wear rate for the “cooler” contacts (ta-C(66) and ta-C(78)). These results thus suggest a minor role played by flash temperature. We added this to the Methods section.

Label	Young's modulus E (GPa)	Density ρ (g cm ⁻³)	sp ³ C p_{sp^3} (%)	Thermal conductivity K (W m ⁻¹ K ⁻¹)	Heat capacity C_p (J g ⁻¹ K ⁻¹)	Thermal diffusivity D (10 ⁻⁶ m ² s ⁻¹)	Maximum flash temperature $T_{fmax, BL}$ (°C)
a-C:H	259	2.07	-	1.01	0.74	0.66	2.3
a-C	287	2.34	0.31	1.12	0.74	0.65	4.1
ta-C(51)	493	2.77	0.62	1.92	0.69	1.01	0.5
ta-C(66)	572	2.91	0.73	2.22	0.67	1.15	2.2
ta-C(78)	625	3.01	0.79	2.43	0.66	1.23	3.1

25. Page 21: Where were the ZDDP and PAO sourced from?

ZDDP and PAO were sourced from Total, France. We added this information to the "Materials" section in Methods.

26. Page 21: It could be worth highlighting that the authors use a primary C₄ ZDDP, which forms a tribofilm much more slowly than secondary ZDDPs (Ref. 15 in the manuscript) and is more thermally stable.[11]

We thank the reviewer for the useful information. We added a sentence about the effect of ZDDP's chemical structure to the "Materials" section in Methods. The impact of the structure of ZDDP additive is under consideration for another study.

27. Page 21: Which DLC hardness does this maximum Hertzian contact pressure correspond to?

The maximum Hertzian contact pressure of 210 MPa is the same for all DLCs. Our DLC coatings are only 2 μm thick, so the macroscopic Hertzian contact pressure is calculated using mechanical properties of the steel substrate. We added this explanation to the "Tribological tests" section in Methods.

28. Page 23: Were the surfaces relaxed before ZDDP molecules were added between them? Otherwise high surface energies are likely, making reactivity larger than would be the case for experimental surfaces.

DLC surfaces were fully relaxed before ZDDP insertion between them. We added this sentence to the "Quantum chemical calculations" section in Methods.

29. Supporting Information: Typo – title of reference 3 should be: 'A hybrid Gaussian and plane wave density functional scheme'

We thank the reviewer for pointing out this typo. We corrected it in Supplementary References.

References

- Neville, A., Morina, A., Haque, T. & Voong, Q. Compatibility between tribological surfaces and lubricant additives - How friction and wear reduction can be controlled by surface/lube synergies. *Tribol. Int.* **40**, 1680–1695 (2007).
- Spikes, H. Friction Modifier Additives. *Tribol. Lett.* **60**, 5 (2015).
- Ueda, M., Kadirci, A. & Spikes, H. ZDDP Tribofilm Formation on Non-Ferrous Surfaces. *Tribol. Online* **15**, 318–331 (2020).
- Savvides, N. & Bell, T. J. Microhardness and Young's modulus of diamond and diamondlike carbon films. *J. Appl. Phys.* **72**, 2791–2796 (1992).
- LiBassi, A. et al. Density, sp³ content and internal layering of DLC films by X-ray reflectivity and electron energy loss spectroscopy. *Diam. Relat. Mater.* **9**, 771–776 (2000).
- Taylor, L. J. & Spikes, H. A. Friction-enhancing properties of ZDDP antiwear additive: Part I-Friction and morphology of ZDDP reaction films. *Tribol. Trans.* **46**, 303–309 (2003).
- Archard, J. F. Contact and Rubbing of Flat Surfaces. *J. Appl. Phys.* **24**, 981 (1953).
- Liu, J. et al. Tribochemical Wear of Diamond-Like Carbon-Coated Atomic Force Microscope Tips. *ACS Appl. Mater. Interfaces* **9**, 35341–35348 (2017).

9. Schall, J. D., Gao, G. T. & Harrison, J. A. Effects of Adhesion and Transfer Film Formation on the Tribology of Self-Mated DLC Contacts. *J. Phys. Chem. C* 114, 5321–5330 (2010).
10. Archard, J. F. The temperature of rubbing surfaces. *Wear* 2, 438–455 (1959).
11. Jones, R. B. & Coy, R. C. The Chemistry of the Thermal Degradation of Zinc Dialkyldithiophosphate Additives. *ASLE Trans.* 24, 91–97 (1981).

Thank you for suggesting the interesting papers. We carefully read the papers, and cited some of them in the revised manuscript.

REVIEWERS' COMMENTS

Reviewer #1 (Remarks to the Author):

This is a review for the revision. The authors well revised the manuscript, reflecting the reviewers' suggestions and comments. Questions have been well answered and clarified. Although the technical scope of the results seems to be adequate for lubrication/wear specialists, the approaches and findings of the research can be of interest to a broad general audience, as it shows state of the art in its investigation. The paper is recommended for publication.

Reviewer #2 (Remarks to the Author):

Review of revised manuscript number NCOMMS-20-49225A entitled "Interplay of mechanics and chemistry governs wear of diamond-like carbon coatings interacting with ZDDP-additivated lubricants" that has been submitted for publication in Nature Communications.

The authors have done a sterling job at addressing the extensive comments raised by the reviewers. I am still not convinced that this paper qualifies as being of general interest to the scientific community as it does address a fairly specialized problem, and the same issue was raised by Reviewer #3. However, the journal editor seems to have decided that it is.

It is still also rather long and somewhat dense reading and the addition comments added in response to the reviewers' comments have not helped.

I still have my doubts about the validity of simulations for correctly describing such interfacial phenomena but the inclusion of additional analyses of the formation of subsurface sulfur has strengthened the connection.

These comments notwithstanding, the manuscript can now be accepted for publication.

Reviewer #3 (Remarks to the Author):

The authors have addressed all the issues raised in my original review, so I am happy with this version of the MS.